# Memory Efficient Continual Learning with Transformers

**Beyza Ermis**
Amazon Web Services
ermibeyz@amazon.com

**Giovanni Zappella**
Amazon Web Services
zappella@amazon.com

**Martin Wistuba**
Amazon Web Services
marwistu@amazon.com

**Aditya Rawal**
Amazon Web Services
adirawal@amazon.com

**Cédric Archambeau**
Amazon Web Services
cedrica@amazon.com

## Abstract

In many real-world scenarios, data to train machine learning models becomes available over time. Unfortunately, these models struggle to continually learn new concepts without forgetting what has been learnt in the past. This phenomenon is known as catastrophic forgetting and it is difficult to prevent due to practical constraints. For instance, the amount of data that can be stored or the computational resources that can be used might be limited. Moreover, applications increasingly rely on large pre-trained neural networks, such as pre-trained Transformers, since the resources or data might not be available in sufficiently large quantities to practitioners to train the model from scratch. In this paper, we devise a method to incrementally train a model on a sequence of tasks using pre-trained Transformers and extending them with Adapters. Different than the existing approaches, our method is able to scale to a large number of tasks without significant overhead and allows sharing information across tasks. On both image and text classification tasks, we empirically demonstrate that our method maintains a good predictive performance without retraining the model or increasing the number of model parameters over time. The resulting model is also significantly faster at inference time compared to Adapter-based state-of-the-art methods.

## 1 Introduction

Transformers [57], e.g. BERT [12], have shown their effectiveness in various natural language processing (NLP) tasks such as classification [25], Natural Language Inference [41, 39], and Question Answering [16]. Inspired by this achievement, some pioneering works have recently been introduced on adapting Transformers architectures to Computer Vision (CV). Vision Transformers [13, 55] showed that a pure Transformer applied directly to a sequence of image patches can perform well on image classification tasks. Besides, some recent studies [67, 4, 30] showed that Transformers are generalized to new domains given only a few samples. Transformers show a great ability to learn complex concepts but when confronted with a sequence of different tasks they tend to "overwrite" the previously learnt concepts. In general, deep networks suffer heavily from this phenomenon, called *catastrophic forgetting* (CF) [35], impeding continual or lifelong learning. In the last few years, a growing body of works attempted to tackle CF in continual learning (CL) [14, 19, 27, 46, 63, 65] but most of them are not able to meet the scale or accuracy requirements of real-world applications. Moreover, adapting large-scale pre-trained Transformer models to downstream tasks via fine-tuning is the method of choice in NLP applications, posing the need for methods that can directly work with pre-trained models instead of requiring the training of a new model from scratch [15].

36th Conference on Neural Information Processing Systems (NeurIPS 2022).

In this work, we tackle both text and image classification problems in a setting where the number of tags or classes associated to the input data grows over time. In fact, the ability to continually extend the set of tags or classes used to categorize the content is a major problem in many applications. For example, newspapers can tag news according to topics of interest such as "sport", "politics", "food" by using a pre-trained language model and refining it using a few hundred pre-tagged articles. New tags may appear over time, for example "COVID-19" was a completely unknown news category in 2019 but appeared frequently since it is emerged. In these cases, retraining models from scratch is often impractical and can lead to inconsistencies in the labeling when compared to the one provided by the previous model. In particular, we focus on incrementally extending classifiers based on pre-trained Transformer models given their ubiquity in NLP and the growing interest in CV.

To address the issue of incremental fine-tuning of pre-trained Transformers in the sequential learning setting without CF, we propose Adaptive Distillation of Adapters (ADA). ADA leverages Adapters [20], a specialized neural network module that adds new parameters to the neural network and a distillation mechanism to consolidate the new information with the previously learnt knowledge in a fixed amount of parameters with little amount of forgetting. This method allows the user to control the memory consumption, while retaining state-of-the-art performance when running the algorithm on sequences of tens of tasks. This tight memory control is important in industrial applications. The alternative, a model growing in size with the number of tasks, would require a change in hardware to adapt to the growing memory requirements of the deployed model. This would be problematic since a practitioner will incur into higher risk of system instability and be forced to make conservative hardware choices.

The main contribution of our work is ADA, an algorithm that can achieve high predictive performance on both text and image classification in different continual learning scenarios. ADA also provides lower inference time and uses an order of magnitude fewer parameters than state of the art methods such AdaptersFusion [39]. Additionally, we implemented Adapters for vision Transformers and empirically demonstrated their effectiveness.

## 2 Related Work

Adapters [20] were proposed for fine-tuning of pre-trained language models and were studied for the multi-task setting. AdapterFusion [39] provides state-of-the-art performance by composing the pre-trained Adapters and it can simply be repurposed for preventing CF in CL by learning one Adapter for every new task. While it has been shown that the number of additional model parameters per Adapter is significantly smaller than the number of parameters used in the pre-trained model [20] (e.g., $3.6\%$ of the parameters of the pre-trained model), since both Adapters and AdapterFusion require to store all the model parameters, the memory consumption increases rapidly with the number of tasks. In the case of a model being trained on 30 tasks, we would have to add more parameters than the number of pre-trained Transformer parameters (details in Section 5). The linear increase in memory and storage consumption making the method unsuitable for CL.

Recent work studied catastrophic forgetting [53, 10, 25, 33] and incremental learning [64] for NLP and CV [30, 15]. Pasanuru et al. [38] focus on the few-shot setting where only a few data points are available for each task. Ke et al. [26] proposed an architecture to achieve both CF prevention and knowledge transfer. This method has some similarity to AdapterBERT [20] since they insert a CL plug-in module in two locations in BERT. A CL-plugin is a capsule network [50] that uses one separate capsule [18] (2-layer fully connected network) for each task, and like Adapters, memory increases linearly over the time. In addition, this algorithm requires to learn task masks to address knowledge transfer, which is costly to compute. Among those recent works, only a few [30, 15] have applied the Transformers architecture to CL on image datasets. In [30], for each new task, the model is copied and fixed to be used as the teacher model in the distillation phase. The student model is trained on both new task samples together with the knowledge distillation loss that uses samples from old tasks which is stored in the rehearsal memory. In [15], the authors aim to learn a unified model that will classify an increasingly growing number of classes by building upon a new architecture. However, they need to train a new Transformer, where the process is very costly and contrast with our goal of using public pre-trained models. To the best of our knowledge there is no method able to leverage public pre-trained Transformers while keeping the number of model parameters constant and retaining state-of-the-art predictive performance.

# 3 Problem Setup and Preliminaries

**Problem Setup.** A sequence of classification tasks $\{T_1, \ldots, T_N\}$ are given where each task $T_i$ contains a different set of data sample (text or image)-label training pairs $(x_{1:t}^i, y_{1:t}^i)$ and contains $c$ new classes namely $Y_i = \{Y_i^1, \ldots, Y_i^c\}$ with $t$ examples for each new class. The goal of the learner is to learn a set of parameters $\tilde{\Theta}$ such that $\frac{1}{N} \sum_{i \in \{1, \ldots, N\}} \text{loss}(T_i; \tilde{\Theta})$ is minimized. The task identifier is provided to the learner with every new batch of data. Moreover, in our specific case, $\tilde{\Theta}$ is composed of a set of parameters $\Theta$ provided by a pre-trained model and, depending on the algorithm, some additional parameters which need to be learned for each specific task. In its simplest case, this additional set of model parameters can just be a head model, but some algorithms use significantly more elaborate functions. In the case of the labeling application described in Section 1, each task represents a label and the learner creates a new binary classifier for each label.

For the training of task $T_i$, the learner can only access the newly added examples and label names in this task. To evaluate the learner, the test data consists of examples across all the previous tasks, where the potential label space for the test example is $Y_1^{1:c} \cup Y_2^{1:c} \cup \cdots \cup Y_N^{1:c}$. All methods that we define in the following sections receive as input a pre-trained model $f_\Theta(.)$, e.g., BERT [12], that is able to extract high quality representations from the input data.

**Adapters.** Adapters were proposed by [20] as an alternative to fine-tuning in NLP. They add new modules between layers of a pre-trained network called *Adapters*. These modules are feed-forward layers that project the original feature size to a smaller dimension and projects them to the original size thereafter, ensuring that the number of parameters stays substantially small as compared to the original model. (See Appendix A.4 for the details of the Adapter architectures.) Adapters share the pre-trained model parameters $\Theta$ across all tasks and introduce a small number of task-specific parameters $\Phi_i$ without affecting previous ones. The model is initialized with parameters of a pre-trained model $\Theta$. For each of the task $i \in \{1, \ldots, N\}$ where $N$ is the total number of tasks, a set of new and randomly initialized Adapter parameters $\Phi_i$ are introduced. The parameters $\Theta$ are fixed and only the parameters $\Phi_i$ are trained when a new task is added. This makes it possible to train Adapters for all $N$ tasks, and store the corresponding knowledge in designated parts of the model. The objective for each task $i \in \{1, \ldots, N\}$ is of the form: $\Phi_i \leftarrow \arg\min_\Phi L_i(D_i; \Theta, \Phi)$.

AdapterFusion [39], has been proposed to mitigate the lack of knowledge sharing across tasks. It works in two phases: i) in the knowledge extraction stage, adapters, which encapsulate the task-specific information, are learnt for each of the $N$ tasks; while ii) in the knowledge composition stage, the set of $N$ Adapters are combined by using additional parameters $\Psi$. The additional parameters $\Psi_i$ for task $i$ are defined as: $\Psi_i \leftarrow \arg\min_\Psi L_i(D_i; \Theta, \Phi_1, \ldots, \Phi_i, \Psi)$. While this provides good predictive performance, in the CL setting, new tasks are added sequentially and storing a large set of Adapters $\Phi_1, \ldots, \Phi_N$ is practically infeasible.

# 4 Adaptive Distillation of Adapters (ADA)

To address the issues we mentioned in the previous sections, we propose Adaptive Distillation of Adapters (ADA). ADA keeps a fixed amount of Adapters in memory and takes transferability of representations into account to effectively consolidate newly created Adapters with previously created ones. ADA works in two steps: i) it trains a new Adapter and classification head, which we refer as the *new* model, using the training dataset of the new task; ii) it consolidates the *old* model with the new model. To better control the memory usage, ADA has a fix budget for the number of Adapters $K$ that are stored in a pool of old models. In the consolidation phase, the algorithm selects one of the models in the pool using scores that quantify the transferable information contained in the representations they provide. In the following sections, we explain the components of ADA and how they work.

## 4.1 Distillation of Adapters

For each new task $T_n$, the Adapter parameters $\Phi_n$ are added to the model, while the pre-trained model parameters $\Theta$ are kept frozen and are never changed. Only the task-specific Adapter parameters $\Phi_n$ and the head model parameters $h_n$ are trained for the current task. The model $f_n(x; \Theta, \Phi_n, h_n)$, with parameters $\Theta$, $\Phi_n$ and $h_n$ is called the *new* model. The head model parameters are fixed after training

the new model and they are not updated during model consolidation. When a prediction for a task $T_i$ is required, the corresponding Adapter $\Phi_{\gamma(i)}$ and head model $h_i$ is called. $\gamma$ is a mapping from the task id to the corresponding Adapter in the pool or to the newly trained Adapter. We abuse notation defining $f$ as the function returning the output (*logits*) on all tasks:

$$f(x; \Theta, \Phi, h) = \left[ f(x; \Theta, \Phi_{\gamma(1)}, h_1), \ldots, f(x; \Theta, \Phi_{\gamma(n-1)}, h_{n-1}), f(x; \Theta, \Phi_{\gamma(n)}, h_n) \right] \quad (1)$$

For the consolidation step, an Adapter from the pool $\Phi$ is selected as explained in Section 4.2 and new collection of Adapter pool $\Phi'$ is created where the selected Adapter is replaced with $\Phi_c$. $\Phi_c$ denotes the consolidated model parameters and preliminarily the parameters are randomly initialized. Similarly, a copy of $\gamma$ is created to map the old tasks that are associated to the selected Adapter and the new task $n$ to $\Phi_c$. The consolidation then has the following objective:

$$\min_{\Phi_c} \frac{1}{|\mathcal{D}_{distill}|} \sum_{i=1}^{|\mathcal{D}_{distill}|} (f(x_i; \Theta, \Phi, h) - f(x_i; \Theta, \Phi', h))^2 \quad (2)$$

where $\mathcal{D}_{distill}$ denotes the unlabeled training data used for distillation, and the distillation loss is computed as the difference between the logits produced by the existing specialist models denoted by $f(x_i; \Theta, \Phi, h)$ and the consolidated model denoted by $f(x_i; \Theta, \Phi', h)$ based on $L_2$ loss. After $\Phi_c$ has been trained, $\Phi$ is swapped with $\Phi'$. This is a high-level view of the mechanism, our implementation is optimized to avoid copying models when not necessary.

This procedure follows the double distillation loss [69] which is originally proposed for class incremental learning to train a new Adapter that is used with the pre-trained model to classify both old and new tasks. The main idea is first training a separate model for the new class(es) using labeled data, and then combining the new and old models using unlabeled distillation data via a double distillation training objective. We generalize this solution to our case where we have a set of teacher models kept in Adapter pool $\Phi$ and train a student model $\Phi_c$. Double distillation procedure and the alternative solutions for distillation are discussed in Appendix A.1 but this solution was the best performing one in our experiments.

While several different data sources can be used to populate the buffer, such as using auxiliary external data [69] or generating synthetic data [9], in this work we populate the buffer using covariates from previous tasks selected with Reservoir Sampling [58]. This simple mechanism may not be the most effective, but it will guarantee that no advantage is given to ADA in the experimental comparison.

## 4.2 Adapter Selection for Distillation

In the previous section, we assumed the Adapter to be consolidated as given but ADA keeps a pool of Adapters and the selection of the Adapter to be distilled is an important part of the algorithm. In fact, our empirical observations show that a random selection of the Adapter provides poor performance (see Section 5.3). The intuition behind our selection mechanism is the following: since a specialized head for every task is created, we can assume that when the features provided by the associated Adapter are highly informative, the updates (i.e., the gradients applied) will be small. At the same time, training a new head with every Adapter in the pool in order to observe which one is the most effective would increase the amount of computation required and significantly impact the usability of the method. The problem of computing the information carried by a representation in an efficient manner has been already studied in the transfer learning community [3, 56, 54].

While, the aim of that research is completely different and, to the best of our knowledge, there is no clear relation between transferability and forgetting, the mathematical foundation of this work are closely related to our intuition. In fact, scores like TransRate [22] employ the mutual information between the features provided by a pre-trained model and the target labels for the task at hand. When the mutual information is high, the transferability is high. More specifically, the knowledge transfer from a source task $T_s$ to a target task $T_t$ is measured as:

$$\text{TrR}_{T_s \to T_t}(f(\Theta, \Phi_{\gamma(s)})) = H(Z) - H(Z|Y), \quad (3)$$

where $Y$ are the labels of target examples and $Z = f(X; \Theta, \Phi_{\gamma(s)})$ are features of them extracted by the pre-trained model and the Adapter associated to the source task.

TransRate is not the only score designed to quantify transferability between a pre-trained model and a new task: *Log Expected Empirical Prediction* (LEEP) [36] is a well-known alternative. Also in

this case, the score was designed with a different application in mind, but it leverages the conditional distribution of the target label given the source label to quantify the how informative the information provided by the source model is. Specifically, LEEP is a three steps method. At Step 1, it computes dummy label distributions of the inputs $f(X; \Theta, \Phi_{\gamma(t)}, h_t)$ in the target data set $\mathcal{D}$. At Step 2, it computes the empirical conditional distribution $\hat{P}(y|z)$ of target label $y$ given the source label $z$. At Step 3, it computes LEEP using $f(X; \Theta, \Phi_{\gamma(s)}, h_s)$ and $\hat{P}(y|z)$:

$$L(f(\Theta, \Phi_{\gamma(s)}, h_s), \mathcal{D}) = \frac{1}{m} \sum_{i=1}^{m} \log \left( \sum_{z \in \mathcal{Z}} \hat{P}(y|z) f(X; \Theta, \Phi_{\gamma(s)}, h_s)_z \right), \qquad (4)$$

where $z$ is a dummy label randomly drawn from $f(X; \Theta, \Phi_{\gamma(s)}, h_s)$ and $y$ is randomly drawn from $\hat{P}(y|z)$. We selected TransRate and LEEP for their simplicity and their ability to provide a quantification without training but practitioners can replace these scores with different ones as they see fit.

### 4.3 Algorithm

---

**Algorithm 1** Adaptive Distillation of Adapters (ADA)

---

**Require:** $\Theta$: pre-trained model, $K$: adapters pool size
  Freeze $\Theta$ and create $\gamma = Map()$
  **for** $n \leftarrow 1$ to $N$ **do**
    A task $T_n$ is received
    Initialize $\Phi_n$
    Process $T_n$, train new model $f_n(x; \Theta, \Phi_n, h_n)$
    Sample from $T_n$ and add to distillation data $\mathcal{D}_{distill}$
    **if** $n \leq K$ **then**
      Store $f_n$ in the pool
    **else**
      $j^* \leftarrow \arg\max_{j \in \{1, ..., K\}} \text{TranScore}(T_n, f_j)$
      Add $(n, j^*)$ to $\gamma$
      Consolidate model:
        $f_{j^*} = \text{Distill}(f_{j^*}, f_n, \mathcal{D}_{distill})$
    **end if**
    Serve predictions for any task $i \leq n$ using $f_{\gamma(i)}$
  **end for**
  $\text{Distill}(f_i, f_j, \mathcal{D}_{distill})$:
    Get soft targets $\hat{y}_i$ from old model $f_i$ with $\mathcal{D}_{distill}$
    Get soft targets $\hat{y}_j$ from new model $f_j$ with $\mathcal{D}_{distill}$
    Initialize $\Phi_c$
    Compute distillation loss and train model $f(x; \Theta, \Phi_c)$ as defined in Eq. 2
    **return** $f$

---

ADA is detailed in Algorithm 1. The graphical workflow of the algorithm is shown in Appendix 7. For every new task, the algorithm trains a new adapter and head model (called $\Phi_n$ and $h_n$). If the adapters pool did not reach the maximum size yet (controlled by $K$), it just adds it to the pool. If the pool reached the maximum size, the algorithm is forced to select one of the adapters already in the pool and distill it together with the newly trained one. In order to select which adapter to distill, ADA uses the transferability scores (e.g., LEEP or TransRate). Once the adapter in the pool with the highest transferability score (called $f_{j^*}$) is identified, it consolidates that adapter and the newly trained one into a new adapter and replaces the old one present in the pool. In order to be able to make effective predictions, the algorithm also keeps a mapping $\gamma$ of which adapter in the pool must be used in combination with each of the task-specific heads.

## 5 Experiments

In this section, we empirically validate our adapter distillation approach on text and image classification tasks and show that ADA achieves similar performance to AdapterFusion while consuming significantly less memory. We dedicate Section 5.3 to ablation studies providing further insights into the mechanisms implemented in ADA and their contribution.

**Datasets and experimental setup.** We use three text datasets for multi-label text classification: Arxiv Papers [66] (paper classification), Reuters (RCV1-V2) [29] (news classification), Wiki-30K [71] (Wikipedia article classification) and two dataset for image classification: CIFAR100 [28] and MiniImageNet [49]. Details about the datasets are given in Appendix A.3.

For the *multi-label text classification* experiments, we first sample a sequence of labels from the label space. Then, we create a balanced binary classification task for each label by sampling the same amount of positive data points from the label considered and negative data points from the labels preceding the current one in the sequence. After splitting the data in training and test set, we

provide the algorithm with the training set and subsequently measure its performance on the test set. The algorithm never observes any data point in the test set and, more generally, every data point in the dataset is used only once. For Arxiv Papers and Reuters datasets, we created 20 tasks and for Wiki-30K 60. We fixed the number of training samples per task to 100. The test set consists of 40 data points on Reuters and of 100 data points on Arxiv and Wiki-30K.

For *image classification*, we design two scenarios. In the first scenario, each new task is a balanced binary classification problem. Each class can be selected to be the positive class only once. In the second scenario each task is a balanced multi-class classification problem with 5 classes. In both cases we provide the learner with 50 data points per class both at training and test time: in the first scenario each task will have a training set of 250 data points and in the second case of 100 data points. The total number of tasks is fixed to 20 for both scenarios. The distillation memory size is fixed to 1000 for Wiki-30K which has a larger number of tasks, and to 500 for the others.

**Metrics.** In [34], three metrics that we discuss in the following are defined to evaluate the performance of a CL method. We use these metrics to evaluate our methods. It is considered that we have access to a test set for each of the $N$ tasks in $\{T_1, \ldots, T_N\}$. After the model finishes learning about the task $T_i$, its test performance are evaluated on all $N$ tasks. By doing so, a matrix $R \in \mathbb{R}^{N \times N}$ is constructed where $R_{i,j}$ is the test classification accuracy of the model on task $T_j$ after observing the last sample from task $T_i$. Letting $\bar{b}$ be the vector of test accuracies for each task at random initialization, the three metrics are defined: i) Average Accuracy = $\frac{1}{N} \sum_{i=1}^{N} R_{N,i}$, ii) Backward Transfer (BWT) = $\frac{1}{N-1} \sum_{i=1}^{N-1} R_{N,i} - R_{i,i}$ and iii) Forward Transfer (FWT) = $\frac{1}{N-1} \sum_{i=2}^{N} R_{i-1,i} - \bar{b}_i$. (The larger these metrics, the better the model.) All the results in this section are averaged over 5 runs.

**Baselines.** We compare ADA the following baselines. 1) *Fine-tuning head model (B1)*: We freeze the pre-trained representation and only fine-tune the output layer of each classification task. The output layer is multiple-head binary classifer that we also use for the other methods. 2) *Fine tuning the full model (B2)*: We fine-tune both the pre-trained representation and the output layer for each classification task. 3) *Adapters* [20]: We train and keep separate Adapters for each classification task as well as the head models. 4) *AdapterFusion* [39]: It is a two stage learning algorithm that leverages knowledge from multiple tasks by combining the representations from several task Adapters in order to improve the performance on the target task. This follows exactly the solution depicted in Section 3. 5) *Experience Replay (ER)* [48]: ER is a commonly used baseline in Continual Learning that stores a subset of data for each task and then "replays" the old data together with the new one to avoid forgetting old concepts. [11] propose to use such a memory module for sparse experience replay and local adaptation in the language domain. This method stores all training examples, in order to achieve optimal performance. To make this method comparable with *adapter-based* methods, we freeze pre-trained representation, add a single adapter parameters $\Phi$ and train the adapter by replaying examples from old tasks while training using data from the new task. In order to keep baselines comparable we assign to ER the same amount of memory is used for the distillation buffer in ADA. In addition to these baselines, we use one special case of ADA with K=1 as a baseline to demonstrate the advantage of effective consolidation of Adapters.

**Adapter architectures.** We use pre-trained models from HuggingFace Transformers [61] as our base feature extractors. We ran experiments with BERT$_{base}$, DistilBERT$_{base}$, RoBERTa$_{base}$ for text classification and ViT-B and DeiT-B for image classification. We analyze the cases based on all these models, due to the space constraints, we present BERT$_{base}$ in this section and the rest in Appendix A.5. BERT$_{base}$ model uses 12 layers of Transformers block with a hidden size of 768 and number of self-attention heads as 12 and has around 110 M (440 MB) trainable parameters. For the Adapter implementation, we use Adapter-Hub [40], but no Adapter implementation was available for Vision Transformers. We define our architecture of Adapters for ViT and DeiT in Appendix A.4. An Adapter has a simple bottleneck architecture that contains fewer parameters than the attention and the feed-forward layers. The Adapter size is the hyper-parameter that is tuned and it can be set to $\{12, 24, 48, 96, 192, 384\}$ for BERT$_{base}$ model. For all the methods, we use the same configuration for the Adapters, setting the size to $48$. With this setting, an Adapter contains $\sim 1.8$ M parameters. We also train a head model for each task, that has 768 parameters for BERT$_{base}$ (last hidden size of BERT$_{base} \times$ output size, which equals to 1 for binary classification). The tables in Appendix A.5 reports the number of parameters used for baselines and ADA in our experiments.

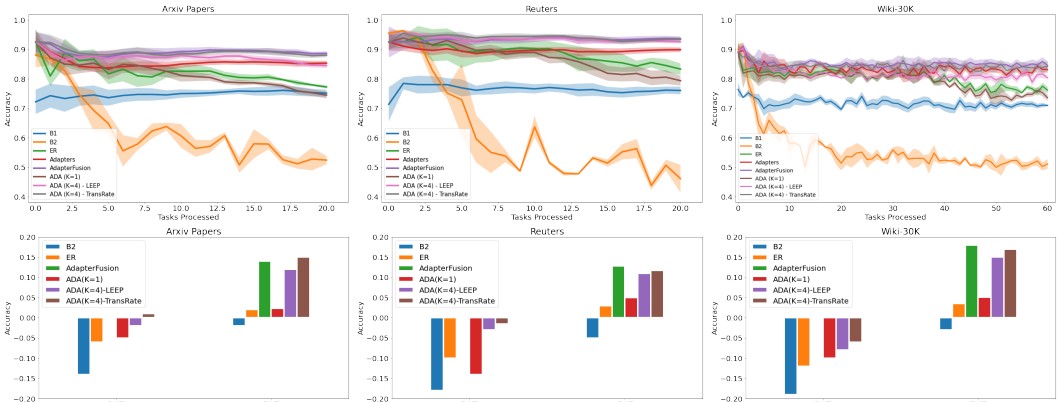

Figure 1: Comparison between baselines and ADA on Arxiv, Reuters and Wiki-30K. On top, we report the number of tasks processed on the x-axis and we report the average accuracy measured on the test set of the tasks processed on the y-axis, shaded area shows standard deviation. On bottom, we report BWT and FWT.

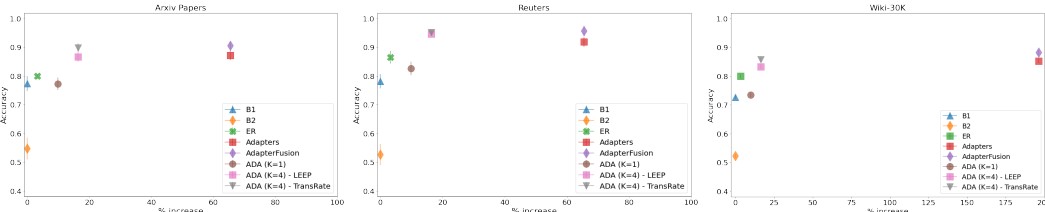

Figure 2: Comparison of the % increase in the number of parameters of baseline methods and ADA on Arxiv, Reuters and Wiki-30K. The predictive performance reported on the y-axis is measured after processing all tasks.

## 5.1 Text Classification

**Predictive performance.** Figure 1 shows the comparison of ADA and the baseline methods. It can be clearly seen that freezing all pre-trained model parameters, and fine-tuning only the head models (B1) led to an inferior performance compared to adapter-based approaches. The main reason is that the head models have small amount of parameters to train and fine-tuning only the heads suffers from under-fitting. B2 performs good only for first 2-3 tasks, since we keep training the complete model, it forgets the previously learned tasks very quickly. As mentioned above, Adapters and AdapterFusion add $\sim 1.8$ M parameters for each task and train these parameters with new task data, and these parameters are fixed after training. So, they perform well on both new tasks and previous tasks. The results on each dataset confirm this. Both ER and ADA K=1, perform closely with Adapters almost for half of the tasks. The similar behavior of ER and ADA K=1 demonstrates that the distillation with soft labels works well and it is almost as good as training with the true labels. Later the performance declines for both methods, because the capacity of the Adapter is exceeded. ADA LEEP and ADA TransRate results with K=4 Adapters show that selective consolidation of Adapters significantly improves the performance. Their performance is on par with AdapterFusion while the number of model parameters is significantly lower. We present $\text{BERT}_{base}$ results in this section while the rest is reported in Appendix A.9.

We also compute FWT and BWT scores for these methods. We didn't present B1 and Adapters in the plots, since both FWT and BWT are zero for them. BWT is zero for AdapterFusion, since the fusion parameter is computed with available Adapters, and the Adapters trained later is not used for the previous tasks. ADA-LEEP and ADA-TransRate minimizes negative backward transfer, while showing a positive forward transfer for all datasets.

**Memory consumption.** Figure 2 shows the increase in terms of percentage in the number of parameters used by each method and their predictive performance. We see that on Wikipedia, 200% of the base model parameters ( $220M$ additional parameters) are added. These results make clear that

ADA is significantly more efficient in terms of memory usage. It can achieve predictive performance similar to the one of Adapters and AdapterFusion while requiring significantly less model parameters. On Reuters and Arxiv, it can store the parameters of only 5 Adapters (K=4 Adapters in the pool, and one Adapter for new task), against the 20 required by AdapterFusion (on Wikipedia it is 5 against 60).

**Inference time.** When machine learning models are used to power customer-facing web sites, they are often required to provide predictions in a few milliseconds to keep the overall latency within requirements. Moreover, in this kind of application the model will be trained once and make billions of predictions so a reasonable increase in the training time is irrelevant compared to a decrease in the inference time. We report the inference time results of ADA and other Adapter based methods in Appendix A.6. Results demonstrate that ADA provides a sufficiently fast inference for most applications and still offers opportunities to speed it up further, for example by employing smaller pre-trained Transformers (e.g. DistilBERT, see Appendix A.5).

**Training time.** Distillation of Adapters brings an extra cost for ADA while learning fusion parameters brings an extra cost for AdapterFusion. Computing transferability takes constant time which is negligible. Distillation costs training an additional Adapter ( 1.6 % of full fine-tuning time of BERT). Figure 10d in AppendixA.6 reports the average training time comparison on Wiki-30K that is the largest difference with AdapterFusion given larger number of tasks. We can clearly see that the difference is small (ADAis 3.37% more, ADA-TransRate is 5.6% more) while the difference between the inference time is significant.

## 5.2   Image Classification

For image classification experiments, we add *Elastic Weight Consolidation (EWC)* [27] as an additional baseline since it is widely used in CL literature for image classification. EWC is a regularization-based CL method that assumes that some weights of the trained neural network are more important for previously learned tasks than others. During training of the neural network on a new task, changes to the weights of the network are made less likely the greater their importance.

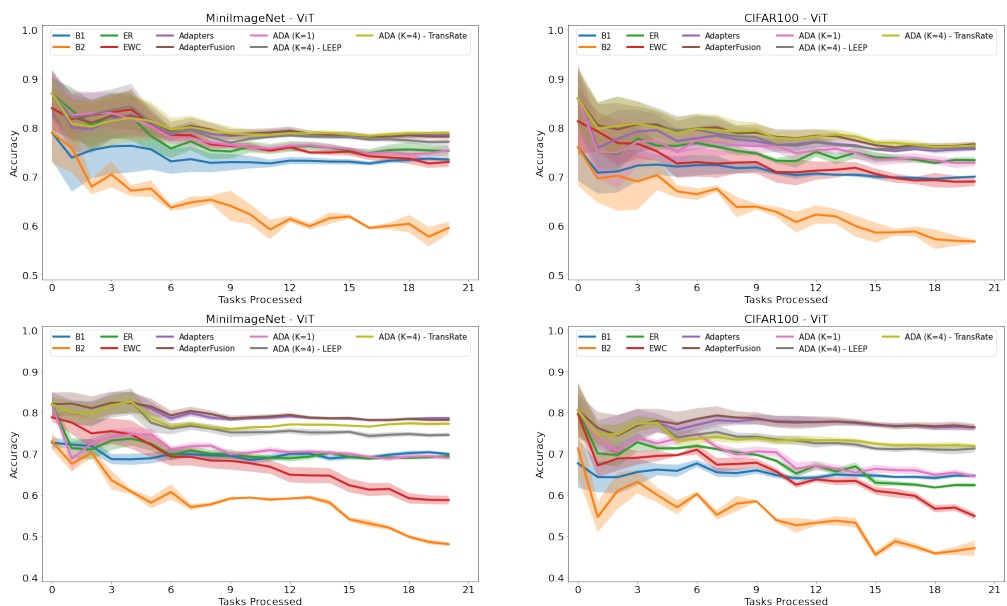

Figure 3: Comparison between baselines and ADA with ViT model on MiniImageNet and CIFAR100. Top figures shows the binary, and bottom figures shows the multi-class classification results.

Figure 3 shows the comparison of ADA and the baseline methods. The results show the same behaviour with text classification. B1 leaded to an inferior performance compared to other approaches. B2 performs well only for initial tasks and it forgets the previously learned tasks very quickly. Results confirm there is no forgetting for Adapters and AdapterFusion. Although the careful tuning of regularization coefficient, EWC cannot handle CF, especially for multi-class classification problem. ADA with K=1 shows that distillation alone doesn't prevent forgetting. In almost all cases, ER

performs on-par with ADA K=1, providing evidence that a small amount of memory can actually improve performance compared to fine-tuning or regularization, but the improvement is limited and does not last as the number of tasks increases.

ADA-LEEP and ADA-TransRate results with K=4 Adapters show that selective consolidation of Adapters significantly improves the performance. For binary classification, their performance are on par with AdapterFusion while the number of model parameters is significantly lower. For multi-class, their performance slightly declines after a certain number of tasks. This is discussed in next section and the main reason is that the capacity of the Adapter is exceeded. To validate the interoperability of ADA to different models, we run the same experiments on DeiT model and present the results in Appendix A.10 due to space constraints.

### 5.3 Ablation studies

**Comparison with larger distilled models.** In Section 5 we compared ADA with the special case of ADA with K=1 to evaluate the improvement provided by our approach over a distillation-only solutions. We would like to provide additional observations of the superior performance of ADA by comparing its performance with the one of a distilled Adapter using more parameters. Specifically, we run an experiment where we compare ADA with K=4 and ADA with K=1 as displayed before but in this case the "size" of the Adapter, which is 48 for Size$\times$1, is multiplied by 4 for Size$\times$4 Adapter to have a comparison where the different methods use the same number of model parameters. Since K=1 is a special case where a single Adapter is kept in the pool, the transferability metric is irrelevant and we can see ADA with K=1 as a method purely based on distillation like DMC [69].

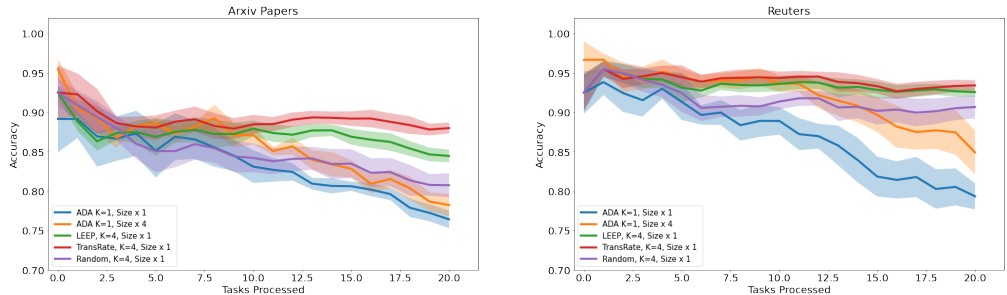

Figure 4: Impact of *LEEP* and *TransRate* when the total number of Adapter parameters is same on Arxiv and Reuters.

The results reported in Figure 4 show that ADA can make a better use of the model parameters compared to a distillation-only method and that the intelligent selection of which Adapters to distill together makes once again a big difference. It is also interesting to observe that the usage of additional model parameters brings a clear advantage but the mixed comparison between the ADA K=4 with random Adapter selection and ADA K=1 with four times larger Adapters leaves some questions open regarding how far distillation can get in this setting. Another finding is that *TransRate* outperforms *LEEP* in most cases. It is also demonstrated in the original paper [22] that *TransRate* has a strong correlation to the transfer learning performance and it outperforms *LEEP* and other metrics employed.

**Impact of the Adapters pool size.** In our experiments we used a fixed number of Adapters in the pool size, but more Adapters can be added to ADA's pool as more tasks are processed. This may actually be the preferred usage in some applications. We already know that having an Adapter per task provides good performance and using multiple of them at the same time like in AdapterFusion provides a benefit, but we would like to verify the sensitivity to this parameter. The results reported in Figure 5 show a rapidly decreasing added value when the number of Adapters grows, a behavior which aligns well with our practical requirements of keeping the number of model parameters under control when the number of tasks increases. See additional experiments in Appendix A.11.

**Mixed Data Experiments.** We run experiments in a setting where we sample 200 tasks from Arxiv, Reuters and Wikipedia (50/50/100) respectively (the order of the tasks are created randomly). We fixed the number of training and test samples per task to 100. Figure 6 shows that we observe a little saturation only after the 150th task when $K$=4 and no saturation when $K$=8. Besides, ADA with *TransRate* comparable performance with Adapters and AdapterFusion even in a complicated setting.

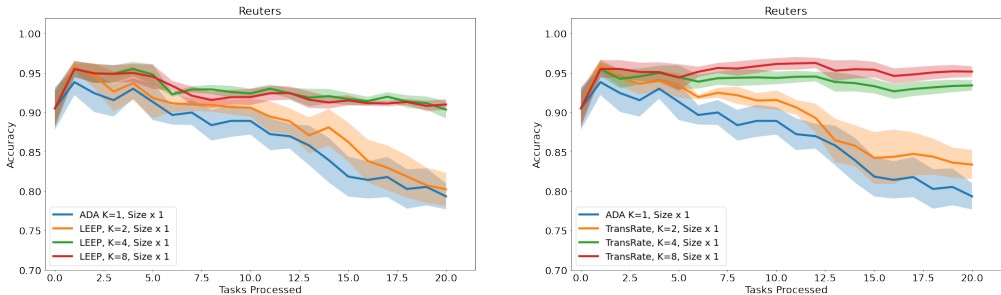

Figure 5: *LEEP* and *TransRate* performances when $K = \{1, 2, 4, 8\}$ on Reuters.

Figure 6 also shows the increase in terms of percentage in the number of parameters used by each method and their predictive performance. We see that $\sim 330\%$ of the base model parameters are added for Adapters and AdapterFusion. These results make clear that ADA is significantly more efficient in terms of memory usage while keeping the comparable performance.

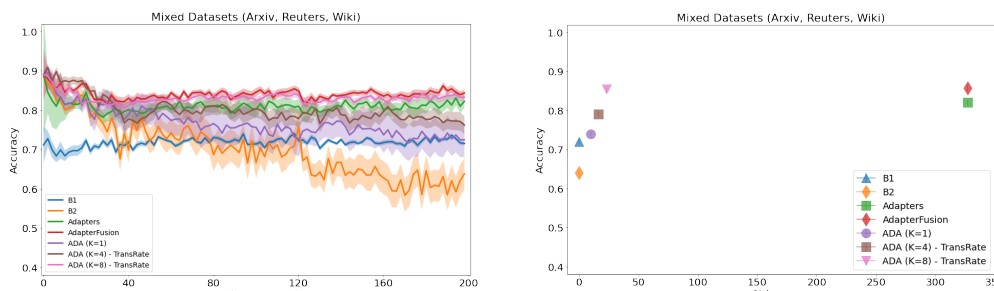

Figure 6: (Left) method performance comparisons (Right) comparison of the $\%$ increase in the number of parameters of baseline methods and ADA on mix of datasets. *LEEP* shows a similar performance with *TransRate*, for the sake of clarity, we didn't add it to the figures.

# 6 Conclusion

In this paper we presented ADA, a method that allows neural text and image classifiers to learn new classes based on pre-trained Transformers while maintaining strict control of the memory usage and reaching state-of-the-art predictive performance. The method has shown to be effective in different domains and allows users to leverage publicly available pre-trained Transformers for continual classification tasks. We evaluated ADA on different classification tasks and demonstrated that the predictive performance is competitive with state-of-the-art methods which use up to an order of magnitude parameters. ADA also displayed lower latency at inference time and improved data efficiency for some specific settings (see Appendix A.8). Moreover, we empirically demonstrated that Adapters can give good results when used in combination with vision Transformers on CV tasks.

Transformers are very popular, but they are not the only models being widely used in practice. We consider this the main weakness of our approach and we would like to further expand our activity to perform CL on other widely used pre-trained models such as ResNet. Addressing multi-modal classification using text and images together will be the other focus of our future research.

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
