# A  Memory Efficient Continual Learning with Transformers: Appendix

## A.1  Related work on CL approaches, distillation and transferability

**Continual Learning (CL).**  Existing methods for CL can be roughly categorized as follows: (1) Replay-based methods [34, 48, 11, 8, 60] retain some training data of old tasks and use them in learning a new task to circumvent the issue of catastrophic forgetting (CF); (2) Regularization-based methods [27, 2, 23] add a regularization term to the loss to consolidate previous knowledge when learning a new task; (3) Gradient-based methods [68, 1] ensure the gradient updates occur only in the orthogonal direction to the input of old tasks and thus will not affect old tasks. Recently, some studies use pre-trained models for class incremental learning [26, 21]; (4) Parameter isolation-based methods [25, 62] allocate model parameters dedicated to different tasks and mask them out when learning a new task; (5) Meta-learning-based methods, which directly optimize the knowledge transfer among tasks [47, 37] or learn robust data representations [24, 60].

**Distillation for CL.**  Knowledge distillation refers to the process of transferring the knowledge from a large bulky model or a set of models to a single smaller model that can be practically deployed under real-world constraints. Essentially, it is a form of model compression that was first proposed by [5] and used by [17] to preserve the output of a complex ensemble of networks when adopting a simpler network for more efficient deployment. The idea is adopted in CL and incremental learning domain to maintain the responses of the network unchanged on the old tasks whilst updating it with new training samples in different ways [52, 7, 31, 70]. [52] propose an end-to-end learning framework where the representation and the classifier are learned jointly without storing any of the original training samples. [31] distill previous knowledge directly from the last trained model. [70] propose to use the current model to distill knowledge from all previous model snapshots, of which a pruned version is saved. [51] use distillation to consolidate the network after each task has been learned and [6] leverage knowledge distillation for retaining past experience.

We inspired from the idea proposed by Zhang et al. [69] where two individual image classification models trained on image data of two distinct set of classes (old classes and new classes) are consolidated into one single model that can classify all classes. The training objective for consolidation is defined as:

$$\min_{\Theta} \frac{1}{\mathcal{U}} \sum_{x_i \in \mathcal{U}} L_{dd}(\boldsymbol{y}_i, \hat{\boldsymbol{y}}_i) \tag{5}$$

where $\mathcal{U}$ denotes the unlabeled auxiliary training data and the double distillation loss $L_{dd}$ is defined as:

$$L_{dd}(\boldsymbol{y}_i, \hat{\boldsymbol{y}}_i) = \frac{1}{t} \sum_{j=1}^{t} (y^j - \hat{y}^j)^2 \tag{6}$$

in which $y^j$ is the logit produced by the consolidated model for the $j$-th class. In our work, we adopt the idea of model consolidation and use it for incremental text classification. In our setting, we leverage the pre-trained model, keep it fixed, and only use Adapters to transfer knowledge from old tasks to the new tasks and train one Adapter that can perform well on all classification tasks. Our main goal is to use the advantage of knowledge transfer between tasks with distillation. So we also use transferability estimation methods to select the Adapters that needs to be distilled. By enhancing the power of distillation, we achieve the same performance with state-of-the-art methods while keeping the number of model parameters much smaller.

**Task Transferability.**  Automatically selecting intermediate tasks that yield transfer gains is critical when considering the increasing availability of tasks and models. There are a number of works that explores task transferability in NLP [42, 32, 59, 45, 44]. Poth et al. [43] present a large-scale study on Adapter-based sequential fine-tuning. Given multiple source and target task pairs $(s,t)$, they first train an Adapter on $s$, then fine-tune the trained Adapter on $t$ and show the relative transfer gains across the different combinations. They use different methods for intermediate task selection, and LEEP [36] is one of the methods that they used in this work to measure transferability and it is consolidated in NLP domain. TransRate [22] is a very recent work and it is used with image classification tasks in the original work. To the best of our knowledge, we use TransRate for the first time in NLP domain. Our work is quite different from what is proposed in the literature. We focus on selecting the best representation from a pool of representations (trained Adapters) for model consolidation, without the necessity of computationally expensive additional approach. We use *proxy estimators*, LEEP and TransRate, that evaluate the transferability of pre-trained models towards a target task without explicit training on all potential candidates.

## A.2  ADA algorithm

The visual representation of ADA is shown in Figure 7.

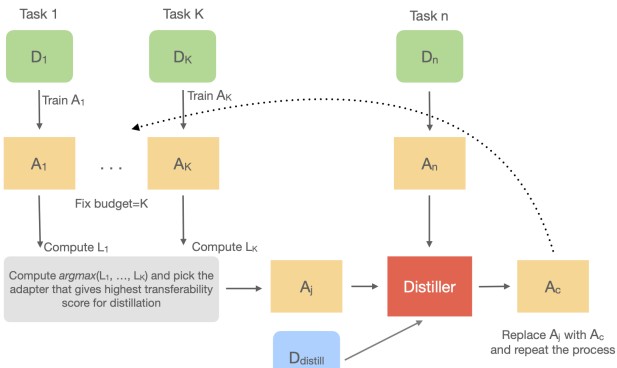

Figure 7: ADA workflow.

## A.3 Datasets and Experimental Setup

**Datasets.** Arxiv papers dataset contains the abstract and the corresponding subjects of 55,840 papers in the computer science field from Arxiv.org. There are 54 subjects in total and each paper can cover multiple subjects. In our work each of these subjects will represent a different task for the classifier where the target is to predict corresponding subjects of an academic paper according to the content of the abstract. Reuters consists of over 800,000 manually categorized newswire stories made available by Reuters Ltd for research purposes. Multiple topics can be assigned to each newswire story and there are 103 topics in total. For Wiki-30K, a set of tags for the English Wikipedia was gathered. Starting with a set of more than 2 million articles from the English Wikipedia on April 2009, the tag information for each of these articles was retrieved from the social bookmarking site Delicious. Only the articles annotated by at least 10 users in Delicious were preserved. As a result, a dataset with 20,764 tagged Wikipedia articles was generated. There are 29,947 labels in this dataset. Both CIFAR100 [28] and MiniImageNet [49] consist of 60000 colour images in 100 classes, with 600 images per class.

**Setup.** We use *Adam* as optimizer with the batch size of 8. For learning rate, we select best from $\{0.00005, 0.0001, 0.0005, 0.001\}$ after observing the results on the first five tasks.
We tune the regularization coefficient of EWC by grid search in $\{0, 1, 10, 100, 1000\}$.

As computation environment, we used Amazon G4dn instances that provide up to 8 NVIDIA T4 GPUs, 96 vCPUs, 100 Gbps networking, and 1.8 TB local NVMe-based SSD storage and are also available as bare metal instances.

## A.4 Adapter Architecture

**Architecture.** Figure below shows the Adapter architecture and it's integration with transformer. In [20], they add the adapter module twice to each Transformer layer: after the projection following multi-headed attention and after the two feed-forward layers. To limit the number of parameters, a bottleneck architecture is proposed. The adapters first project the original $d$-dimensional features into a smaller dimension, $m$, apply a non-linearity, then project back to $d$ dimensions. The total number of parameters added per layer, including biases, is $2md + d + m$. By setting $m \ll d$, the number of parameters added per task is limited.

**Vision Adapters.** One other contribution of this work is using Adapters approach with vision Transformers for the first time on sequential image classification tasks, validating that Adapters work with vision Transformers and show that ADA can achieve predictive performance on-par with AdapterFusion. We implement vision Transformer Adapters in AdapterHub [40][1] (that has Apache License, Version 2.0). As in AdapterBERT [20], we insert a 2-layer fully-connected network in each Transformer layer of ViT [13] and DeiT [55] is built upon the ViT architecture, so an Adapter is added in the same way.

## A.5 Trainable parameters for different models

The tables below reports the number of parameters used for baselines and ADA in our experiments. We reported all the cases for different models: $BERT_{base}$, $RoBERTa_{base}$ and $DistilBERT_{base}$. We don't add the head size to the table, since it's very small and same for all the methods.

---

[1] https://github.com/Adapter-Hub/adapter-transformers

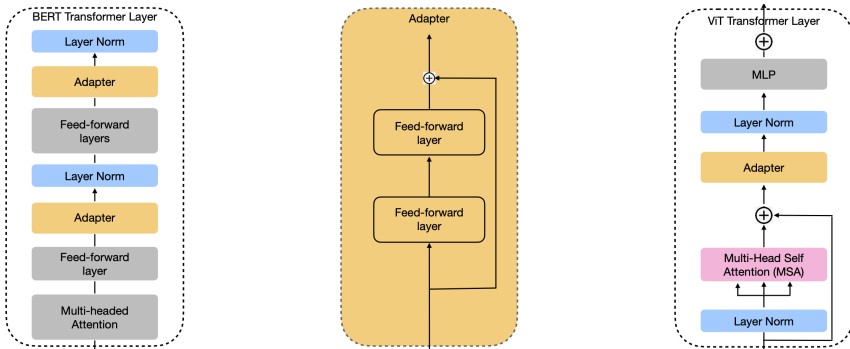

Figure 8: Left shows AdapterBERT [20] in a BERT transformer layer, and middle shows the Adapter architecture. Right shows our Adapter implementation in a ViT [13] transformer layer. As in AdapterBERT, we added an Adapter before layer norm and feed-forward layers (MLP).

Table 1: The number of all parameters and those used for training and inference as well as the model size of methods for $\text{BERT}_{base}$. $K$ is the number of Adapters in the pool, and $F$ is the number of fused Adapters (it is between 2 and number of tasks). For the *Adapters* $F = 1$.

| | Fine-Tuning | | | |
|---|---|---|---|---|
| | Trainable | Inference | Total | Total (Size) |
| Task = $\{1, 10, 30, 60\}$ | 110 M | 110 M | 110 M | 440 MB |

| | Adapters & AdapterFusion | | | |
|---|---|---|---|---|
| | Trainable | Inference | Total | Total (Size) |
| Task = 1 | 1.8 M | 111.8 M | 111.8 M | 447.2 MB |
| Task = 10 | 1.8 M | 110 + (F×1.8) M | 128 M | 512 MB |
| Task = 30 | 1.8 M | 110 + (F×1.8) M | 164 M | 656 MB |
| Task = 60 | 1.8 M | 110 + (F×1.8) M | 218 M | 872 MB |

| | ADA | | | |
|---|---|---|---|---|
| | Trainable | Inference | Total | Total (Size) |
| Task = 1 | 1.8 M | 111.8 M | 111.8 M | 447.2 MB |
| Task = $\{10, 30, 60\}$ | 2×1.8 M | 111.8 M | 110 + (K+1)×1.8 M | 440 + (K+1)×7.2 MB |

Table 2: The number of all parameters and those used for training and inference as well as the model size of methods for $\text{RoBERTa}_{base}$.

| | Fine-Tuning | | | |
|---|---|---|---|---|
| | Trainable | Inference | Total | Total (Size) |
| Task = $\{1, 10, 30, 60\}$ | 125 M | 125 M | 125 M | 500 MB |

| | Adapters & AdapterFusion | | | |
|---|---|---|---|---|
| | Trainable | Inference | Total | Total (Size) |
| Task = 1 | 1.8 M | 126.8 M | 126.8 M | 507.2 MB |
| Task = 10 | 1.8 M | 125 + (F×1.8) M | 143 M | 584 MB |
| Task = 30 | 1.8 M | 125 + (F×1.8) M | 179 M | 716 MB |
| Task = 60 | 1.8 M | 125 + (F×1.8) M | 233 M | 932 MB |

| | ADA | | | |
|---|---|---|---|---|
| | Trainable | Inference | Total | Total (Size) |
| Task = 1 | 1.8 M | 126.8 M | 126.8 M | 507.2 MB |
| Task = $\{10, 30, 60\}$ | 2×1.8 M | 126.8 M | 125 + (K+1)×1.8 M | 500 + (K+1)×7.2 MB |

Table 3: The number of all parameters and those used for training and inference as well as the model size of methods for DistilBERT$_{base}$.

| | Fine-Tuning | | | |
|---|---|---|---|---|
| | Trainable | Inference | Total | Total (Size) |
| Task = $\{1, 10, 30, 60\}$ | 66 M | 66 M | 66 M | 264 MB |

| | Adapters & AdapterFusion | | | |
|---|---|---|---|---|
| | Trainable | Inference | Total | Total (Size) |
| Task = 1 | 0.9 M | 66.9 M | 66.9 M | 267.6 MB |
| Task = 10 | 0.9 M | 66 + (F×0.9) M | 75 M | 300 MB |
| Task = 30 | 0.9 M | 66 + (F×0.9) M | 93 M | 372 MB |
| Task = 60 | 0.9 M | 66 + (F×0.9) M | 120 M | 480 MB |

| | ADA | | | |
|---|---|---|---|---|
| | Trainable | Inference | Total | Total (Size) |
| Task = 1 | 0.9 M | 66.9 M | 66.9 M | 267.6 MB |
| Task = $\{10, 30, 60\}$ | 2×0.9 M | 66.9 M | 66 + (K+1)×0.9 M | 264 + (K+1)×3.6 MB |

Table 4: The number of all parameters and those used for training and inference as well as the model size of methods for ViT-B (Same for Deit-B). $K$ is the number of Adapters in the pool, and $F$ is the number of fused Adapters (it is between 2 and number of tasks). For the *Adapters* $F = 1$. For ER, for Task = $\{1, 10, 20\}$, it is same with ADA Task=1. Total size is in MB.

| | Fine-Tuning (B1, B2) and EWC | | | |
|---|---|---|---|---|
| | Trainable | Inference | Total | Total (Size) |
| Task = $\{1, 10, 20\}$ | 86 M | 86 M | 86 M | 344 |

| | Adapters & AdapterFusion | | | |
|---|---|---|---|---|
| | Trainable | Inference | Total | Total (Size) |
| Task = 1 | 1.8 M | 87.8 M | 87.8 M | 351.2 |
| Task = 10 | 1.8 M | 86 + (F×1.8) M | 104 M | 416 |
| Task = 20 | 1.8 M | 86 + (F×1.8) M | 122 M | 488 |

| | ADA | | | |
|---|---|---|---|---|
| | Trainable | Inference | Total | Total (Size) |
| Task = 1 | 1.8 M | 87.8 M | 87.8 M | 351.2 |
| Task = 10 | 2×1.8 M | 87.8 M | 86 + (K+1)×1.8 M | 344 + (K+1)×7.2 |
| Task = 20 | 2×1.8 M | 87.8 M | 86 + (K+1)×1.8 M | 344 + (K+1)×7.2 |

Table 4 reports the number of parameters used for baselines and ADA in image classification experiments with ViT and DeiT. We don't add the head size to the table, since it's very small: 768 parameters per binary head, 15K parameters (6 KB) for 20 tasks, 3840 per multi-class head, 75K parameters (30KB) for 20 tasks. Also they are same for all the methods.

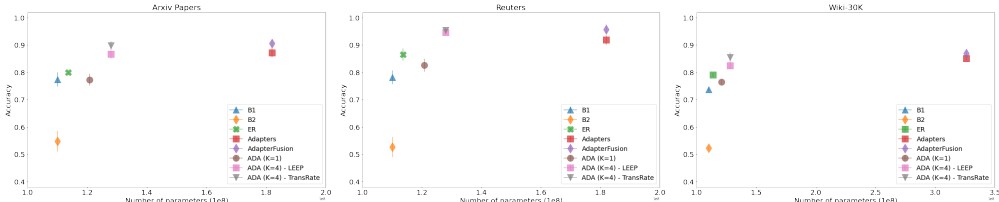

Figure 9: Comparison of number of parameters of baselines and ADA on Arxiv, Reuters and Wiki-30K. The predictive performance reported on the y-axis is measured after processing all tasks.

Figure 9 and Table 5 show the number of parameters used by each method and their predictive performance. These results make clear that ADA is significantly more efficient in terms of memory usage. It can achieve predictive performance similar to the one of Adapters and AdapterFusion while requiring significantly less model parameters. On Reuters and Arxiv, it can store the parameters of only 5 Adapters (K=4 Adapters in the pool, and one Adapter for new task), against the 20 required by AdapterFusion.

Table 5: The accuracy (with standard deviation) reported after last task and number of total parameters (Num Params) kept in memory for Adapters, AdapterFusion and ADA variants for experiments on Arxiv, Reuters, Wiki-30K and Mixed setting experiments with BERT$_{base}$.

| | Arxiv | | Reuters | | Wiki-30K | | Mixed | |
|---|---|---|---|---|---|---|---|---|
| | Accuracy | Num Params | Accuracy | Num Params | Accuracy | Num Params | Accuracy | Num Params |
| Adapters | $0.872 \pm 0.017$ | 182 M | $0.918 \pm 0.015$ | 182 M | $0.847 \pm 0.006$ | 326 M | $0.820 \pm 0.008$ | 470 M |
| AdapterFusion | $0.905 \pm 0.013$ | 182 M | $0.957 \pm 0.012$ | 182 M | $0.867 \pm 0.009$ | 326 M | $0.833 \pm 0.010$ | 470 M |
| ADA (K=1) | $0.772 \pm 0.021$ | 120.8 M | $0.825 \pm 0.018$ | 120.8 M | $0.770 \pm 0.019$ | 120.8 M | $0.729 \pm 0.017$ | 120.8 M |
| ADA (K=4) - LEEP | $0.867 \pm 0.017$ | 128 M | $0.947 \pm 0.015$ | 128 M | $0.842 \pm 0.013$ | 128 M | $0.795 \pm 0.012$ | 128 M |
| ADA (K=4) -TansRate | $0.898 \pm 0.014$ | 128 M | $0.951 \pm 0.013$ | 128 M | $0.858 \pm 0.011$ | 128 M | $0.812 \pm 0.012$ | 128 M |

## A.6 Inference and training time

In Figure 10a, 10b and 10c we report the average time per prediction made during our experiments. We observe a significant speedup at inference time compared to AdapterFusion. For example, on Reuters, ADA is 5 times faster than AdapterFusion when both K=1 and K=4 (because it always uses one distilled Adapter for inference that has a fixed size). The inference time of AdapterFusion depends on the number of Adapters fused. Results demonstrate that ADA provides a sufficiently fast inference for all datasets.

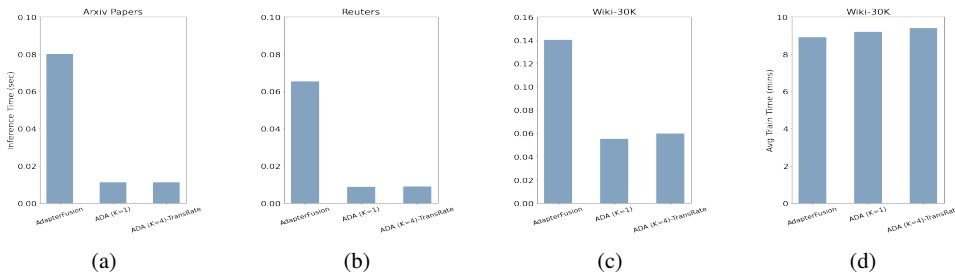

(a)  (b)  (c)  (d)

Figure 10: Comparison of inference times of methods on a) Arxiv, b) Reuters and c) Wiki-30K. d) Comparison of training time on Wiki-30K.

**Training time.** Distillation of Adapters brings an extra cost for ADA while learning fusion parameters brings an extra cost for AdapterFusion. Computing transferability takes constant time which is negligible. Distillation costs training an additional Adapter ( 1.6 % of full fine-tuning time of BERT). Figure 10d reports the average training time comparison on Wiki-30K that is the largest difference with AdapterFusion given larger number of tasks. We can clearly see that the difference is fractional while the difference between the inference time is significant.

## A.7 Memory consumption of ViT and DeiT

Figure 11 shows the number of parameters used by each method and their predictive performance. These results make clear that ADA is significantly more efficient in terms of memory usage also with ViT and DeiT models. It can achieve predictive performance similar to the one of Adapters and AdapterFusion while requiring significantly less model parameters.

## A.8 Additional experiments with different task sizes

We would like to verify if the intelligent distillation mechanism we designed for ADA is not only able to avoid forgetting and save memory but also to increase the data efficiency. Distilling together similar tasks for which a small number of data points is available could also provide a better representation of the data points.

To verify this hypothesis, we repeated our experiments with a variable number of data points in the training set of each task. The amount of positive and negative samples is balanced in both train and test tasks. The size of the training sets of the Reuters tasks contain $t = \{20, 50, 80\}$ samples per class (positive and negative) and the test sets contain 20 samples per class. Arxiv Papers has more samples than Reuters dataset, so we added larger training tasks of size 400 to the configuration, and increased the test task size. For Arxiv, we created the training sets with $t = \{20, 50, 100, 200\}$ samples per positive and negative classes and the test set with 50 samples per class. Our expectation is that by increasing the training set size the overall predictive performance will improve, but we also expect to see the predictive performance of ADA matching (or narrowing the gap with) independent Adapters' one when using a smaller training set.

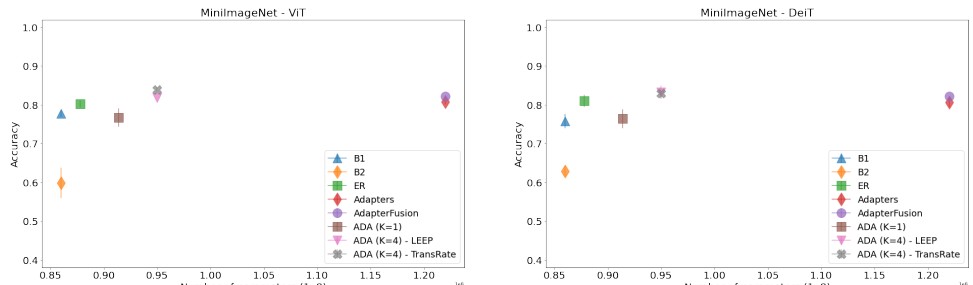

Figure 11: Comparison of number of parameters of baselines and ADA on ImageNet with ViT and DeiT models. The predictive performance reported on the y-axis is measured after processing all tasks.

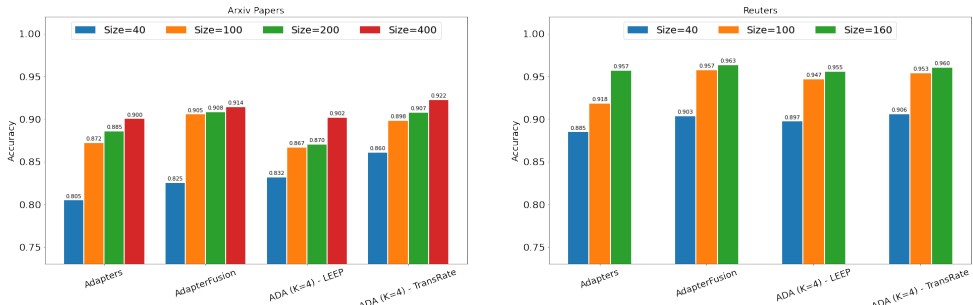

Figure 12: Predictive performance of Adapter based methods with $t = \{20, 50, 100, 200\}$ on Arxiv and $t = \{20, 50, 80\}$ on Reuters.

In Figure 12 we report the results of our experiment. We observe TransRate performing generally better than LEEP, as in previous experiments. Focusing on TransRate, we can see that ADA K=4 with TransRate can actually outperform independent Adapters when the training set size is around 100 data points and even match the performance of independent Adapters using significantly more labels (200 labels on Arxiv and 160 on Reuters). The effect becomes smaller or vanishes when the training set gets larger but this could still bring an important advantage in the "few-shot" setting.

## A.9   Additional experiments with DistilBERT and Roberta

We repeated all the experiments presented in Section 5.1 with DistilBERT$_{base}$ and RoBERTa$_{base}$ as our base models in order to show that it's not only limited to one specific model. The results demonstrated the same trends with BERT$_{base}$ model experiments.

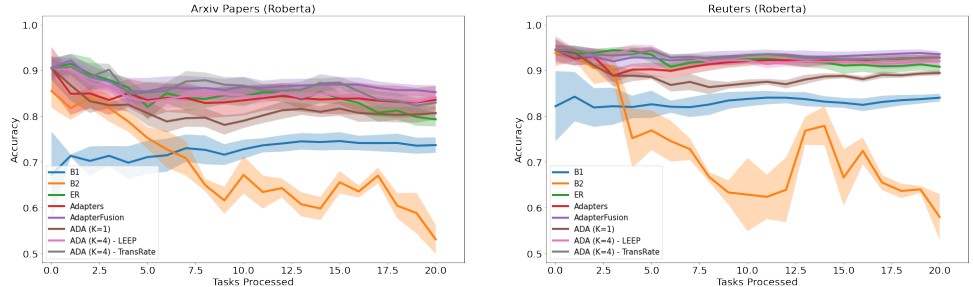

Figure 13: Comparison of baselines and distillation methods on Arxiv and Reuters with RoBERTa$_{base}$. On the x-axis we report the number of tasks processed, on the y-axis we report the average accuracy measured on the test set of the tasks processed, shaded area shows standard deviation.

Figure 13 compares the ADA algorithms with baselines. The findings that we mention in predictive performance is exactly applicable to RoBERTa$_{base}$ results. RoBERTa$_{base}$ performs slightly better on all the methods compared to BERT$_{base}$. The behavior of algorithms are same for DistilBERT$_{base}$ and is very similar to the results with BERT$_{base}$, however, the number of parameters used is different.

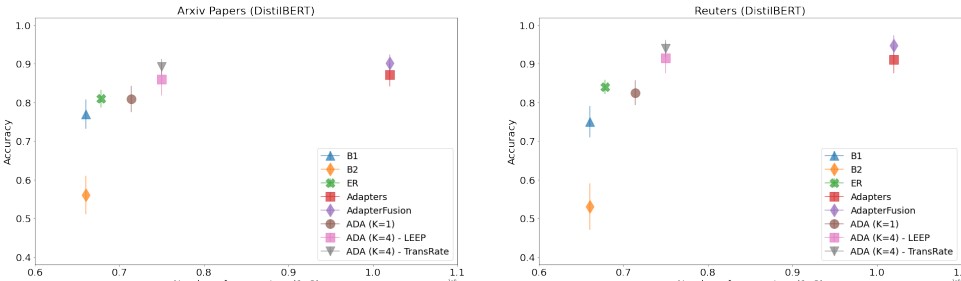

Figure 14: Comparison of number of parameters of baselines and ADA on Arxiv, Reuters and Wikipedia with DistilBERT$_{base}$.

Figure 14 shows the number of parameters used by each method and their predictive performance with DistilBERT$_{base}$ model. (We skip this figure for RoBERTa$_{base}$ because the number of parameters is very close to BERT$_{base}$, and we already show the accuracy in Figure 13.)

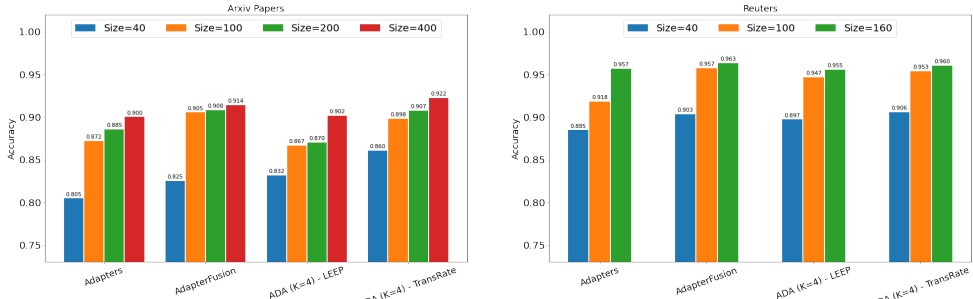

Figure 15: Predictive performance of Adapter based methods with $t = \{20, 50, 100, 200\}$ on Arxiv and $t = \{20, 50, 80\}$ on Reuters.

As in A.8, we report the results of experiments of different task sizes with DistilBERT$_{base}$ in Figure 15. This figure emphasises that with small number of labels and with a model much less parameters, we can still have good prediction accuracy on old and new tasks in CL setting.

## A.10  Additional experiments with DeiT

We repeated all the experiments presented in Section 5.2 with DeiT-B [55] [2] as our base model in order to show that it's not only limited to one specific model. The results demonstrated the same trends with ViT-B [13] [3] experiments.

In Figure 17 we present FWT and BWT scores for baselines. As in Section 5.1, we didn't present B1 and Adapters in the plots, since both FWT and BWT are zero for them. The behaviour is quite similar to text classification experiments. BWT is zero for AdapterFusion, since the fusion parameter is computed with available Adapters, and the Adapters trained later is not used for the previous tasks. ADA-LEEP and ADA-TransRate minimizes negative backward transfer, while showing a positive forward transfer for both MiniImageNet and CIFAR100.

---

[2] https://dl.fbaipublicfiles.com/deit/deit_base_patch16_224-b5f2ef4d.pth
[3] https://huggingface.co/google/vit-base-patch16-224

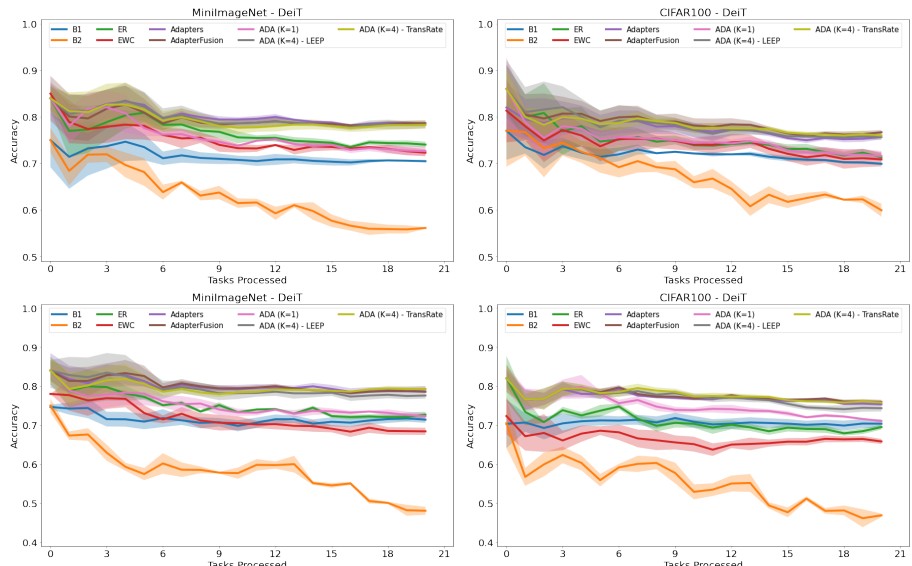

Figure 16: Comparison between baselines and ADA with DeiT model on MiniImageNet and CI-FAR100. Top figures shows the first scenario (binary) results, and bottom figures shows the second scenario (multi-class) results.

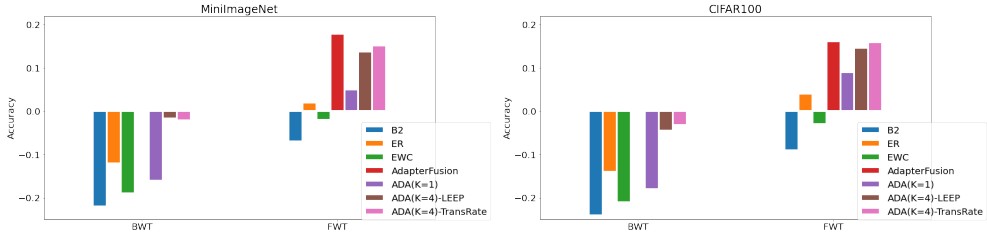

Figure 17: Comparison between baselines and ADA with DeiT model on MiniImageNet and CI-FAR100 for multi-class classification in terms of FWT and BWT.

## A.11 Additional experiments with different Adapters pool size

This section has the additional results with different Adapters pool size on Arxiv Papers dataset. As in Figure 5, the results in Figure 18 show a rapidly decreasing added value when the number of Adapters grows, a behavior which aligns well with our practical requirements of keeping the number of model parameters under control when the number of tasks grows.

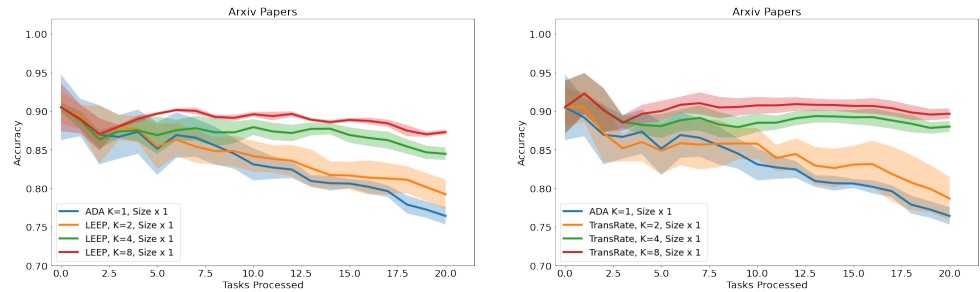

Figure 18: Impact of Adapter pool size for *LEEP* and *TransRate* when $K = \{1, 2, 4, 8\}$ on Arxiv for $t = 50$.