# OpenReview forum: "Memory Efficient Continual Learning with Transformers"
_NeurIPS.cc/2022/Conference — NeurIPS 2022 Accept_

### Official Review · Reviewer_rCKB · 2022-07-17

**Rating:** 6
**Confidence:** 3
**Soundness:** 2 fair
**Presentation:** 2 fair
**Contribution:** 2 fair

**Summary:**

This work presents a method of using adapters to fine-tune pretrained models to downstream tasks in a continual fashion. Instead of adding a new adapter for each new task, the authors define a fixed set of adapters of size K which are repurposed. When the number of tasks are less than K, each new task is trained using a new adapter. For each additional task, first, a new adapter is trained, and then its closest counterpart is selected in the set of K adapters (using TransScore or LEEPl; to measure transferability). The selected adapter is then distilled using a subset of data from all previous tasks that the model has been trained on to prevent forgetting.

The authors convert multi-label and multi-task classification tasks to a series of smaller tasks and report improvement over baselines for both language- and image-based pretrained models in terms of the number of parameters as well as accuracy as more tasks are added.

**Questions:**

It would be great to provide results where all data is used together for multi-label classification following the original dataset with a shared adapter (no continual learning). It will give the readers an idea of existing performance gap.

What is the purpose of eq(1)? Indicating both the individually adapted networks and their set as "f" is confusing. Also line 142, what is \mathcal{U}? It would be great if these notations can be made clearer.

**Strengths And Weaknesses:**

Strengths

1. Considerable improvements in memory requirement as the number of adapters remain fixed while the number of tasks increases.
2. Clever use of transferability metrics to reuse adapters.

Weaknesses:

1. For validating the proposed solution, the authors consider synthetic settings: convert a multi-label classification task to multiple binary classification tasks. In such a situation, finding transferable adapters is much easier since all tasks share an underlying objective. While there are improvements, it is not clear whether, in a realistic setting where tasks may not be as related, the improvements will hold, or if the method will even be able to find transferable adapters. Combining the three datasets on text classification for example would be interesting to see.
2. The writing of section 3 is hard to follow. Many notations have been used without clearly indicating what they mean.

---

> ### Author Response · Authors · 2022-08-02
> **The authors consider synthetic settings: convert a multi-label classification task to multiple binary classification tasks. Combining the three datasets on text classification for example would be interesting to see.**
>
> While we agree that an experiment involving multiple datasets will be an interesting addition, we also think that the scenarios we designed are quite realistic. First we run multi-label text classification experiments on real world datasets where multi-label classification is the main application. For example, the articles provided in RCV1 have been written by Reuters journalists in about two years time span and the labels used in the task are from a real categorization of the articles. Each article in this datasets can be tagged with multiple labels.
> Nevertheless, we will try to run the an experiment on a sequence of tasks containing all the tasks created on the text datasets. Unfortunately running such an experiment for multiple times (to compute average performance) is costly so we are not able to provide the result at this point in time. We hope to be able to share the results before the end of the discussion period.
>
> Moreover, we also use datasets for multi-class classification of images (miniImageNet and CIFAR100). The behaviour of the results of these experiments are pretty similar to the multi-label classification on text datasets. So we can say that when the tasks are not related, ADA still performs well.

---

> ### Author Response · Authors · 2022-08-02
> **The writing of section 3 is hard to follow.**
>
> We updated Section 3 to clarify aspects of our work which may not have been explained as clearly as we intended. We hope many aspects are clear in the current version.

---

> ### Author Response · Authors · 2022-08-02
> **Provide results where all data is used together for multi-label classification following the original dataset with a shared adapter.**
>
> Thank you for the suggestion. We thought that training a separate Adapter for each task gives more capacity to the network and higher accuracy. But we will try to run this baseline after completing the experiment with tasks from multiple datasets.

---

> ### Author Response · Authors · 2022-08-02
> **What is the purpose of eq(1)? What is \mathcal{U}?**
>
> f is the function returning the output (logits) on all tasks and Equation 1 shows which components we need to get the output for each task. f gets the pre-trained model, the adapter assigned for a task and the head model and produces the output when the input x is given.
>
> We are sorry that we had some inconsistency in the notation. \mathcal{U} was used in an early draft as a short version of \mathcal{D}_distill and we forgot to replace this instance. We fixed the problem and the notation should now be consistent everywhere.

---

> ### Author Response · Authors · 2022-08-08
> **Follow-up**
>
> Dear Reviewer rCKB,
>
> We are wondering if you had a chance to consider our responses to your initial assessment of our paper. In particular, we have uploaded a revised version to fix the notations and polish the writing of the Section 3, that you commented that was initially hard to follow. Besides, we clarified in the comments below that our setting is realistic and we do not only consider multi-label classification as we also consider multi-class image classification. You can see more results in the link: https://i.postimg.cc/44wZC2jp/Mixed-Results.png where we explained the setting https://openreview.net/forum?id=U07d1Y-x2E&noteId=uP5A20rFyR1 to address reviewer G9iU's question.
>
> We believe our response and updated materials are addressing not only your concerns but also the current concerns of all reviewers. If you agree that the issues you raised have all been addressed, could you please consider to reflect that in the score appropriately? We would be happy to answer any remaining questions. Thanks!

---

> > ### Comment · Reviewer_rCKB · 2022-08-09
> > **Thanks for the responses**
> >
> > Thank you for providing your responses and providing additional experiments. I have updated my score to 6.

---

> > > ### Author Response · Authors · 2022-08-09
> > > **Thanks for evaluation and follow up**
> > >
> > > Thank you so much for re-evaluating our paper and reflecting to the responses. We hope you will like to see and contribute to the ongoing discussion above, especially about the contribution of our paper.

---

### Official Review · Reviewer_s76Y · 2022-07-17

**Rating:** 5
**Confidence:** 3
**Soundness:** 2 fair
**Presentation:** 2 fair
**Contribution:** 2 fair

**Summary:**

This paper extends the work of Houlsby et al. by proposing a method to distill knowledge between adapters in continual learning. The high level idea is to freeze the foundation model's parameters and train task-specific adapters. The mitigate the lack of knowledge sharing between newly learned adapters, the authors propose a distillation method so that adapters can share knowledge and memory does not grow out of bounds.

**Questions:**

What are some example tasks where this memory reduction is necessary? It's outlined in the introduction, but not explicitly stated.

Where exactly is the "distillation" happening? Which equation, for example.

Why is knowledge distillation a motivation, but the performance is worse than Adapters? Doesn't this mean that it's not distilling knowledge correctly?

**Limitations:**

The authors talk about how their work is tied to transformers, but is this true? Can't this be applied for other model types that allow for task heads?

**Strengths And Weaknesses:**

Strengths:

This paper addresses an important problem in the domain of continual learning. There is typically a tradeoff between learning new tasks well and remembering old tasks. This paper tries to close the gap between this tradeoff by learning new tasks but keeping prior learned tasks in the model's memory.

The memory efficient aspect is interesting, especially if there is a scenario where the number of tasks grow in orders of magnitude.

The introduction was well posed, the task setup is nice, and the motivation is clear.



Weaknesses:

The method section was unclear. Without reading the prior papers such as Houlsby et al., it's hard to understand what is being proposed in this paper. I think the methods section could be condensed and clarified.

The exact concept of ``knowledge distillation'' in terms of the method is unclear.  Algorithm 1 contains a Distill() function, but it's not explained in the paper.

The distinction between \phi, \phi^{'}, \phi_{c} is unclear.

Lastly, the results section is not easy to understand. Without reading prior papers, the BWT and FWT metrics are unclear. The figures are somewhat hard to read. Based on the results, it seems like performance is worse using ADA vs vanilla adapters. The only contribution vs adapters is the memory reduction.


Minor:

"This is an high-level" --> "This is a high-level"

---

> ### Author Response · Authors · 2022-08-02
> **The exact concept of ``knowledge distillation'' in terms of the method is unclear.**
>
> The complete Distill() procedure is defined in Algorithm1 and how the distillation occurs is defined in Equation 2. The complete procedure is explained throughout Section 3.1. We will be happy to answer more detailed questions or modify specific parts of the paper if that will help understanding these concepts more easily.

---

> ### Author Response · Authors · 2022-08-02
> **The distinction between \phi, \phi^{'}, \phi_{c} is unclear.**
>
> Thank you for pointing out this. We updated the notation, we hope it is clear now. \phi is the set of Adapters in the pool before model consolidation, \phi^{'} is the set of Adapters after consolidation has started and \phi_{c} denotes the consolidated Adapter model parameters.

---

> ### Author Response · Authors · 2022-08-02
> **The BWT and FWT metrics are unclear.**
>
> Since BWT and FWT are fairly standard in continual learning, we tried to save some space in the main text but as you point out, this was not a good idea. We updated the paper adding a Metrics section with the definition of the metrics to the main text in Section 4 (Experiments).
>
> We will also try to make the figures bigger if space allows in the next version of the paper.

---

> ### Author Response · Authors · 2022-08-02
> **What are some example tasks where this memory reduction is necessary?**
>
> We can say we need that reduction whenever we have more tasks than the adapter pool size K, but in particular we have a lot of tasks. ADA keeps consolidating the newly trained Adapter by using transferability and keeps the number of Adapters in the pool fixed. You can see how efficient ADA is in Figure 2.
>
> That said, tagging of text document is an application where the number of tags (the tasks in our CL scenario) is very large.
> A classic example is the hierarchy of topics for the RCV1 dataset (https://www.jmlr.org/papers/volume5/lewis04a/lewis04a.pdf), a dataset built on newswire stories made available by Reuters.
>
> Another reference point, less academic and more practical, the NYT.com (http://nyt.com/) website has 5 editions (US, International, Canada, Espaniol, Chinese), in the US edition there are 19 top-level categories (e.g., World, Politics, US, NY, Business), each of these top-level category has sub-categories. For example, “Science” has the following sub-categories “Climate”, “Space & Astronomy”, “Health”, “Trilobites”, “Matter”, “Out There” and “Coronavirus Outbreak” while “Real Estate” has “On the Market”, “Mortgage Calculator”, “Living In”, “The Hunt”, “What You Get”, “The High End”. We did not manually count all of them but there are probably 100+ categories and these are only the one visible to the public, more of them may be used for internal purposes. Each of these subcategories contain hundreds or thousands of articles. When the newspaper wants to change the categories and the subcategories (e.g., “is Trilobites still necessary?”), they probably don’t want to re-tag manually hundreds of thousands of articles nor impact the classification of the articles in the categories not impacted by the change.

---

> ### Author Response · Authors · 2022-08-02
> **Where exactly is the "distillation" happening? Which equation, for example?**
>
> We already explained the changes in the comment "Double distillation loss is not explained in the main body of the paper".
>
> Equation 2 and Distill() procedure in Algorithm 1 show how distillation is happening.

---

> ### Author Response · Authors · 2022-08-02
> **Why is knowledge distillation a motivation, but the performance is worse than Adapters? Doesn't this mean that it's not distilling knowledge correctly?**
>
> The performance is on par with Adapters and AdapterFusion when we leverage transferability scores, especially TransRate
> (see Figure 1 for text classification tasks and Figure 3 for image classification tasks). ADA and its distillation procedure provide a significant advantage in terms of the memory consumption by distilling knowledge in a smaller number of parameters as shown in Figure 2 and discussed in Section 4.1, “Memory Consumption” subsection. In addition, we showed the number of parameters and memory consumption of each method in Table1 and Table2 in Appendix A.5.

---

> ### Author Response · Authors · 2022-08-02
> **The authors talk about how their work is tied to transformers, but is this true? Can't this be applied for other model types that allow for task heads?**
>
> As we discussed in the conclusion, we are currently working on extending ADA for other widely used models. We may have been a bit conservative in our limitations statement, but we prefer to avoid speculations on preliminary results.

---

> ### Author Response · Authors · 2022-08-02
> **Concerns about contribution.**
>
> We hope that we have sufficiently addressed your concerns especially on the contribution of the paper and the usage of the Adapters. We are happy to answer any remaining questions.

---

> ### Author Response · Authors · 2022-08-08
> **Follow-up**
>
> Dear Reviewer s76Y,
>
> We are wondering if you had a chance to consider our responses to your initial assessment of our paper. In particular, we have uploaded a revised version and supplementary material to include multiple discussions for intuition on areas that may have previously been unclear, for example the concept of Adapters.
>
> We believe our response and updated materials are addressing not only your concerns but also the current concerns of all reviewers. If you agree that the issues you raised have all been addressed, could you please consider to reflect that in the score appropriately? We would be happy to answer any remaining questions. Thanks!

---

> > ### Comment · Reviewer_s76Y · 2022-08-09
> > **Still unclear of the contributions**
> >
> > Thanks for the comments. The contributions seems a little clearer. If I understand correctly, the main contribution is still just limiting the memory consumption? I'm not completely convinced on the necessity to share weights between adapters, considering the sharing doesn't lead to performance improvements. Why not just use a new adapter for each task?
> >
> > It's also hard to fully interpret all of the graph figures. Why aren't there tables introduced comparing to the original adapters paper?

---

> > > ### Comment · Reviewer_G9iU · 2022-08-09
> > > **Correctly identified the contributions**
> > >
> > > While I am not the author, I believe you have correctly identified the contribution of the paper as reducing the memory impact of adapters on a large-scale continual learning task. Some performance penalty there is expected as compared to adding a new adapter to every task, but I can easily imagine on a long enough time scale, for a large enough number of tasks, reducing the number of adapters used (because of their memory impact) can be important.

---

> > > > ### Comment · Reviewer_s76Y · 2022-08-09
> > > > **Need for adapters as tasks increase.**
> > > >
> > > > My problem is that the need for adapters as the number of tasks increase to where memory becomes a problem isn't well motivated. Do we really need to share adapter parameters across thousands of tasks? Unless I missed something, I don't believe any of the experiments pushed the limits of memory capacity.
> > > >
> > > > It's hard to sell the need for constraining memory, when all of the experiments are on datasets where the original Adapters method can work fine. If the motivation is datasets like RCV1 and nyt.com, why not run experiments on those?
> > > >
> > > > Furthermore, I think it's hard to claim "state-of-the-art" (L382), when the graphs don't clearly show that and there aren't tables with concrete numbers as in the original Adapters paper.

---

> > > ### Author Response · Authors · 2022-08-09
> > > **Clarification of contributions**
> > >
> > > Dear Reviewer s76Y,
> > >
> > > As Reviewer G9iU correctly pointed out, scaling an approach based on adding a new adapter for every task will not be sustainable in the long term. We offer a solution that using almost-constant memory and predictive performance on-par (difference is not statistically significant) with the solution you are proposing when saying “Why not just use a new adapter for each task?”. Since the figures are difficult for you to interpret, we prepared a table offering evidence of that: https://postimg.cc/6TMJm1yN
> > >
> > > Our experiments may not be “push[ing] the limits of memory capacity”, but scaling up the number of tasks by an order of magnitude or two would make impossible for us to compare 8 different methods (including memory-hungry methods like Adapters/AdaptersFusion) and average their results on several runs, as a sound scientific methodology requires in these cases.
> > > Moreover, as displayed in the experiment with tasks from multiple datasets, reaching 100 tasks, the dynamic is not significantly different even if we did not increase the value of K (for the sake of scientific comparison). Additional experiments could be performed to answer specific research questions (as we did for the ones posed by the other reviewers) but in this case we do not see any hypothesis to be tested in an ad-hoc experiment.
> > >
> > > For the records, the experiments performed for this paper already costed more than 40K $ in cloud compute. It would be unfair to define them small-scale and while it is always possible to run a bigger experiment, it is also useful to evaluate the importance of the information acquired with such work.
> > >
> > > We would also like to point out that your current score classifies our papers as “a paper with technical flaws, weak evaluation, inadequate reproducibility and incompletely addressed ethical considerations”, which does not seem to reflect the content of this discussion.

---

> > > > ### Comment · Reviewer_s76Y · 2022-08-09
> > > > **Memory capacity**
> > > >
> > > > If the main contribution of the paper is reducing the memory so that the model can be run on tasks where standard Adapters fail, I think it should be run on those tasks.
> > > >
> > > > I would argue that this does fall under 3 (weak evaluation). I'm willing to increase my score and decrease my confidence as I'm not an expert in this area.

---

> > > > > ### Author Response · Authors · 2022-08-09
> > > > > **Memory capacity**
> > > > >
> > > > > Dear Reviewer s76Y,
> > > > > it is unfortunate that after two weeks of rebuttal we are still unable to get clarity on what would convince you about the quality of our work. The last answers still leaves two important open points:
> > > > > 1. The memory available on a machine is not a fixed quantity. More memory can be added if the financial situation allows, clusters with terabytes of memory can be created eventually. This does not mean that everyone have access to such a setup and it is not clear which memory size would be able to convince you of the quality of our approach (assuming we would have the budget to run such an experiment).
> > > > > 2. It is not clear which information will be provided by an experiment with a larger number of tasks which is not provided in our current 100 tasks experiment. You say that our evaluation is weak but this statement is not supported by any specific scientific consideration.

---

> > > > > > ### Comment · Reviewer_s76Y · 2022-08-09
> > > > > > **Memory capacity**
> > > > > >
> > > > > > You can still run the baselines (Adapters, AdapterFusion) on 100 tasks (it's also not clear which experiment has 100 tasks), which perform better than ADA. The scientific consideration is that the proposed method is motivated by saying that the baselines can't run large task sizes, but the authors also never runs on task sizes that the baselines cannot.
> > > > > >
> > > > > > If the other reviewers don't think this is a concern, I'm willing to change my score.
> > > > > >
> > > > > > Reviewer G9iU made a similar point saying "It would be informative to see the behavior of ADA in the asymptotic case, where the K adapters are saturated and thus there is a downgrade in the performance. As the paper is presented right now, the readers do not get a balanced picture of where it may fail."
> > > > > >
> > > > > > I don't believe this point has been sufficiently addressed.

---

> > > > > > > ### Author Response · Authors · 2022-08-09
> > > > > > > **Memory capacity**
> > > > > > >
> > > > > > > Dear Reviewer s76Y,
> > > > > > > there are several cloud instances which would not be suitable to serve models of the size we compared in our experiments, but larger instances can be bought spending more money. The point is that even after buying larger instances the memory will not be sufficient if the consumption doubles every 30 tasks or so (and we think this is a conservative estimate).
> > > > > > >
> > > > > > > The experiment involving 100 tasks from the three text datasets used in the experimental section is reported here: https://i.postimg.cc/44wZC2jp/Mixed-Results.png . It is important to observe that we kept the pool size K = 4 for comparing with previous results and try to identify a "saturation point" as requested by your colleagues but we could have increased K to 5 with a negligible memory increase and obtain better predictive performance. We hope this experiment will result convincing and we apologize for not bringing it to your attention before.
> > > > > > >
> > > > > > > Once again, we will be happy to provide more information if needed and we hope that the new evidence will convince you of the quality of our approach.

---

> > > > > > > ### Comment · Reviewer_G9iU · 2022-08-09
> > > > > > > **"You can still run the baseline" is not a sound statement**
> > > > > > >
> > > > > > > While I do not want to intrude in this discussion, "You can still run the baseline" is not a sound statement when it is obvious that the baselines have a higher memory complexity compared to the method under discussion. From what I read in the paper, 100 new tasks mean  (5+1)x memory complexity for adding new Adapters. Now, while you _can_ find machines that can fit 6x LLMs, a method that addresses  such memory hungry methods is better than not using any distillation as long as it doesn't significantly hurt performance.

---

> > > > > > > > ### Comment · Reviewer_s76Y · 2022-08-09
> > > > > > > > **Makes sense**
> > > > > > > >
> > > > > > > > Sure, this makes sense. I'm still wondering how much performance is degraded when tasks increase in orders of magnitude, but I'll increase my score.

---

> > > > > > > > > ### Comment · Reviewer_s76Y · 2022-08-09
> > > > > > > > > **100 task experiments**
> > > > > > > > >
> > > > > > > > > To me, this is the convincing experiment, which wasn't even added to the original manuscript or supplementary. I hadn't seen it if not for going through many comment threads.

---

> > > > > > > > > > ### Author Response · Authors · 2022-08-09
> > > > > > > > > > **100 task experiment**
> > > > > > > > > >
> > > > > > > > > > Dear Reviewer s76Y,
> > > > > > > > > > The experiment was not in the original paper since it was requested during the rebuttal phase and we did our best to share the results in a timely manner. We will add the experiment to the next version of the paper.
> > > > > > > > > > We are glad you appreciated our efforts to provide additional evidence and we hope that this will be reflected in your final score.

---

### Official Review · Reviewer_G9iU · 2022-07-20

**Rating:** 6
**Confidence:** 4
**Soundness:** 3 good
**Presentation:** 2 fair
**Contribution:** 3 good

**Summary:**

In this paper, the authors introduce adaptive distillation of adapters, or ADA algorithm, which helps continual learning with transformers be memory efficient and performant at the same time. The work uses Adapters to fine-tune a large, pretrained transformer model to a new task, as opposed to the standard practice of fine-tuning the final layer. However, as Adapters require more parameters compared to a new final layer, the authors introduce ADA to make continual learning more memory efficient. Basically, ADA keeps a fixed stack of K adapters, and when a new Adapter needs to be added to the stack, ADA takes the Adapter that has the most similarity with the new task and distills that with the new task. The authors run experiments on language and vision tasks to show that ADA can perform continual learning well without catastrophic forgetting without using an unbounded number of parameters. Finally, the authors ablate the impact of adapter size and adapter pool size to show their effects on ADA.

**Questions:**

- What is the relationship between K and the number of tasks at which the adapters get saturated? ADA does not show saturating behavior in the experiments in the paper, but intuitively there should be at some point where it does get saturated.
- What is the performance, quantitatively, of the random-adapter-distillation variation of ADA? Please include that baseline in the experiments as well, as it is currently we do not see any quantitative comparison between this version of ADA and the one with random replacement.

**Limitations:**

- The primary limitation of this work is that it is seemingly limited to transformer-based models only, with no clear indication of how it could transfer to other architectures. However, as of now, transformer-based models are enjoying decent success, and thus this limitation is not a major one.
- The experiments are performed on a handful of tasks, on the order of tens of tasks. It would be informative to see the behavior of ADA in the asymptotic case, where the K adapters are saturated and thus there is a downgrade in the performance. As the paper is presented right now, the readers do not get a balanced picture of where it may fail.
- ADA chooses the adapter to distill at every step in a greedy fashion, instead of optimizing over all the tasks seen so far. The paper does not analyze the worst-case behavior caused by this greedy choice of distilled adapter, or how ADA may be impacted by an unfortunate ordering of tasks.

**Strengths And Weaknesses:**

Overall, the strengths of this paper are as follows:
- This work introduces ADA, which is a novel algorithm for adapting pre-trained transformers to continual learning settings without incurring significant memory overhead.
- The algorithm itself is simple with significant practical benefits as well, as shown by the experiments.
- ADA seems to perform comparatively with baselines while keeping to bounded memory, which is also an impressive feat.

However, there are a few places where this paper could improve, especially concerning its method and experiments.
- Firstly, the paper does not introduce the concept of Adapters sufficiently clearly. The way it is explained right now in the paper is insufficient, in my opinion, for someone that is not already familiar with the concept of adapters. A detailed explanation or a figure to explain the concept of adapters could really go a long way for this.
- Similarly, important concepts such as double distillation loss is not explained in the main body of the paper.
- On Figure 2, the X axis could be more informative if the metric was % increase in parameter per task rather than the total number of parameters in the model. The authors talk about this in the sense that 30 adapters roughly equal the size of a pretrained transformers, which I believe communicates the justification for distillation in the first place, and denoting the modified modules' sizes as % size of the base model would also work better here.

---

> ### Author Response · Authors · 2022-08-02
> **Double distillation loss is not explained in the main body of the paper**
>
> Thanks for bringing this to our attention. We updated the methodology and clarified the double distillation loss. Equation 2 shows how double distillation loss is computed in our case.  And the following paragraph discusses how we modified the double distillation loss proposed by Zhang et al, 2020 for Adapter distillation. The details of how they compute the loss is added to the Appendix A.1 (Equations 5 and 6).

---

> > ### Comment · Reviewer_G9iU · 2022-08-08
> > **Thank you for the responses**
> >
> > Dear authors,
> >
> > After reading through your responses, I am more confident in your submission, and thus I have increased my confidence score from a 2 to a 4. While this paper needs significant polish in terms of including sufficient background to let everyone understand the contributions made here, I believe it is a significant enough contribution to the state of the art that should be included in the further considerations for the practitioners in this field.
> >
> > I wish you best of luck through the review process.

---

> > > ### Author Response · Authors · 2022-08-08
> > > **Follow up**
> > >
> > > Thank you for reading our responses carefully. We would like to inform you that we run additional experiments and the link that you can see the results are posted above that can further reduce your concerns. We would be happy to answer any remaining questions.

---

> ### Author Response · Authors · 2022-08-02
> **Show percent increase in Figure 2**
>
> We updated the plot reporting the percentage increase instead of the absolute value. The figure shows the percentage increase is very big by Adapters and AdapterFusion especially when the number of tasks are bigger. For example, on Wikipedia, we observe 60 tasks, that means 60 Adapters are added and that adds ~200% of the base model parameters additionally to the base model.

---

> ### Author Response · Authors · 2022-08-02
> **What is the relationship between K and the number of tasks at which the adapters get saturated?**
>
> This is a very interesting question. We tried to explore it with the experiments on the Wiki-30K dataset, using a sequence of 60 tasks and K=4. The results we obtained do not show signs of saturation in this case.
>
> The risk of reaching a saturation point leading to performance degradation is not particularly concerning for practical applications because it is trivial to add a new adapter to ADA’s Adapters pool (increase K) at any point in time, creating in this way additional capacity.
>
> Nevertheless, the saturation point poses an interesting scientific question for which we would like to provide some insights. Our experiments suggest that the performance degradation will be influenced not only by the number of tasks but also by their heterogeneity. We are currently working on experiments with a larger number of tasks (see answer on mixing multiple datasets in the same experiment) but unfortunately repeating such an experiment multiple times is slow and costly. We hope to be able to provide additional evidence during the discussion period.

---

> ### Author Response · Authors · 2022-08-02
> **What is the performance, quantitatively, of the random-adapter-distillation variation of ADA?**
>
> The results of a baseline randomly selecting the adapter to distill are already available in Figure 4 (the name is Random). The findings show that ADA can make a better use of the model parameters compared to a distillation-only method and that the intelligent selection of which Adapters to distill by using LEEP and TransRate makes a big difference. Both LEEP and TransRate (red and green lines in the figure) outperforms Random selection (purple line) by margin on both Arxiv and Reuters datasets.

---

> ### Author Response · Authors · 2022-08-02
> **This work is seemingly limited to transformer-based models only**
>
> As mentioned in the conclusions, we are currently working on extending ADA for other widely used models.  We may have been too conservative with the statements in the paper, but we prefer to avoid speculations on early stage results.

---

> ### Author Response · Authors · 2022-08-02
> **The experiments are performed on a handful of tasks, on the order of tens of tasks. Behavior of ADA in the asymptotic case would be informative.**
>
> Our experiments ran on 20 and 60 tasks, which is a significantly larger number than many continual learning benchmarks.
> At the same time, we agree that it would be very interesting to push this further and reach an even larger number of tasks to observe the behavior of ADA in these conditions. Unfortunately these experiments are costly (we avg. on multiple runs) and observing an asymptotic behavior may not be possible within the time allowed for the rebuttal/discussion.
>
> As already mentioned in other answers, we are running an experiment with the task from the three text datasets and we hope to be able to show additional results for that case before the end of the discussion period. We also would like to use this experiment to make some observations on the heterogeneity of the tasks.
>
> With a bit more time we will probably be able to run experiments on at least a dataset on hundreds of tasks, results that we will definitely include in the next version of the paper.

---

> > ### Author Response · Authors · 2022-08-08
> > **Experiment results on multiple dataset**
> >
> > We run additional experiments in a setting where we sample 100 tasks (20/20/60) from Arxiv, Reuters and Wikipedia respectively. The order of the tasks are created randomly and the results are the average of 3 runs. The results can be seen here: https://i.postimg.cc/44wZC2jp/Mixed-Results.png. The results show that ADA is robust and shows comparable performance with Adapters and AdapterFusion even in a complicated setting where the tasks are sampled from different datasets. In addition to that even on 100 tasks which is exceptional in Continual Learning papers in the literature, we observe a little saturation only after the 90th task.

---

> ### Author Response · Authors · 2022-08-02
> **ADA chooses the adapter to distill at every step in a greedy fashion, instead of optimizing over all the tasks seen so far.**
>
> Your point is correct and scientifically interesting. We currently focused on realistic scenarios where very little variance across task ordering was observed, making us believe that for real-world scenario the greedy choice is sufficiently robust.
> At the same time, wanting to investigate the matter further, it would be hard to define what a worst-case scenario is in this case. There is an interesting line of work on which tasks induces most forgetting (e.g., Lee at al. “Maslow's Hammer for Catastrophic Forgetting: Node Re-Use vs Node Activation”, ICML2022) but it is not very clear how to efficiently generate such a task in practice. Moreover, the worst case scenario may not be the same for all algorithms (nor all the possible instances of ADA).
>
> In order to quickly provide an intuition, we can create a scenario where a sequence of tasks T_1, .., T_n is given. The subsequent tasks in the sequence maximize the forgetting as explained in the paper above (assuming we find a way to generate them). This scenario will be very difficult for several continual learning algorithms including our baseline B2 but likely not hurt ADA that much because the algorithm will map these tasks on different adapters in the pool. A significantly more difficult sequence of tasks for ADA (but to some extent easier for B2) would be the one repeating the same task exactly K times with different task IDs (K is the size of the adapters pool) and then performing again the same operation with the task inducing most forgetting given the previous one, and so on. This will not allow ADA to take advantage of the dissimilarity of the different adapters in the pool.
>
> Due to these reasons, studying adversarial sequences of tasks can be quite complicated and of limited practical interest. Moreover, given a scenario built in an adversarial manner, it would be easy to provide an improved version of ADA relying on randomization able to mitigate the damages made by an oblivious adversary.

---

### Author Response · Authors · 2022-08-02
**The paper does not introduce the concept of Adapters sufficiently clearly**

Both Reviewer G9iU and Reviewer s76Y bring our attention that the paper does not introduce the concept of Adapters sufficiently clearly. We extended the discussion of Adapters in Section 2 and referred to the detailed discussion and Figure of architectures given in Appendix A.5. In short, Adapters are feed-forward layers between layers of a pre-trained network. The layer size is smaller than the pre-trained networks's feed forward layer size, so they are called as bottleneck architectures. They project the original feature size to a smaller dimension and projects them to the original size thereafter, ensuring that the number of parameters stays substantially small as compared to the original model. Adapters are trained for each task separately. During the training of a new adapter the parameters of previously trained adapters and of the transformer models remain untouched. In our continual learning scenario, task specific Adapter parameters are added and trained for each new task. AdapterFusion further learns how to combine Adapters for a new task, that mostly increases accuracy while bringing the same increase per task in terms of memory consumption with Adapters.

---

### Author Response · Authors · 2022-08-02
**General response**

We would like to thank the reviewers for their constructive feedback. We updated methodology section to clarify aspects of our work which may not have been explained as clearly as we intended. We also updated Experiments section for a better understanding of some aspects and added some more details to Appendix. We also fixed the typos and other minor problems.

---

### Meta-Review · Area_Chair_AjsK · 2022-08-29

**Recommendation:** Accept
**Confidence:** Certain

**Metareview:**

Some of the reviewers had concerns with experiments that would have compared not distilling (i.e. high-memory-usage everyone gets their own adapter) with the authors' method.  I am willing to recommend acceptance; but I agree that some sort of experiment like this would be useful to contextualize what is traded off for smaller memory footprint.  I would ask the authors to include some experiment like this for the camera ready.

**Award:**

No

---

### Decision · Program_Chairs · 2022-09-14

Accept